# Manipulating 3D Molecules in a Fixed-Dimensional E(3)-Equivariant Latent Space

**Zitao Chen**[1,2*]    **Yinjun Jia**[1*†]    **Zitong Tian**[1,3*]    **Wei-Ying Ma**[1]    **Yanyan Lan**[1,4,5†]

[1] Institute for AI Industry Research (AIR), Tsinghua University
[2] Department of Computer Science and Technology, Tsinghua University
[3] Qiuzhen College, Tsinghua University
[4] Beijing Frontier Research Center for Biological Structure, Tsinghua University
[5] Beijing Academy of Artificial Intelligence
{chen-zt23,tzt23}@mails.tsinghua.edu.cn
{jiayinjun, maweiying, lanyanyan}@air.tsinghua.edu.cn

## Abstract

Medicinal chemists often optimize drugs considering their 3D structures and designing structurally distinct molecules that retain key features, such as shapes, pharmacophores, or chemical properties. Previous deep learning approaches address this through supervised tasks like molecule inpainting or property-guided optimization. In this work, we propose a flexible zero-shot molecule manipulation method by navigating in a shared latent space of 3D molecules. We introduce a Variational AutoEncoder (VAE) for 3D molecules, named MolFLAE, which learns a fixed-dimensional, E(3)-equivariant latent space independent of atom counts. MolFLAE encodes 3D molecules using an E(3)-equivariant neural network into fixed number of latent nodes, distinguished by learned embeddings. The latent space is regularized, and molecular structures are reconstructed via a Bayesian Flow Network (BFN) conditioned on the encoder's latent output. MolFLAE achieves competitive performance on standard unconditional 3D molecule generation benchmarks. Moreover, the latent space of MolFLAE enables zero-shot molecule manipulation, including atom number editing, structure reconstruction, and coordinated latent interpolation for both structure and properties. We further demonstrate our approach on a drug optimization task for the human glucocorticoid receptor, generating molecules with improved hydrophilicity while preserving key interactions, under computational evaluations. These results highlight the flexibility, robustness, and real-world utility of our method, opening new avenues for molecule editing and optimization. [3]

## 1 Introduction

Structure-guided molecule optimization is a crucial task in drug discovery. Medicinal chemists edit molecular structures to improve binding affinity, selectivity, and ADMET (absorption, distribution, metabolism, excretion, and toxicity) properties. These modifications can range from subtle changes, such as introducing a chlorine [1] or methyl [2], to more extensive transformations like deconstruction and reconstruction of known ligands [3] or designing chimera molecules that combine beneficial

---

[*]Equal contribution.
[†]Corresponding author.
[3]The code is available at https://github.com/MuZhao2333/MolFLAE

features of different scaffolds [4, 5]. These diversified tasks present exciting opportunities for deep learning models to accelerate real-world drug design.

Previous generative approaches typically decompose 3D molecule editing into a set of narrowly defined subtasks. Notable progress has been made in molecular inpainting [6, 7, 8], property-guided optimization [9, 10], and shape-conditioned regeneration [11]. While effective, these models often rely on task-specific supervision and architectures, limiting their flexibility and generalizability. Moreover, not all molecule editing tasks align well with the supervised learning paradigm. For example, adding substituents may be too trivial to justify training a specialized model, while complex tasks like scaffold hopping by integrating known actives are often data-scarce for supervised approaches. These limitations call for a more flexible framework capable of supporting a broad spectrum of molecule editing tasks in a unified, data-efficient manner.

Previous successes in image editing and style transfer [12, 13, 14] show that latent space navigation allows for powerful, general-purpose manipulations by perturbing latent vectors. However, 3D molecule generation presents unique challenges not encountered in image domains. Molecules consist of variable numbers of atoms, and they exhibit permutation invariance to the atom order and SE(3)-equivariance to the spatial translation and rotation. These characteristics make latent space modeling significantly more challenging, and most existing 3D generative models operate on the product of latent spaces of each atom or functional group [15, 16, 17, 18, 19], resulting in variable dimensional representations. This variability prohibits common operations on vectors, such as interpolation or extrapolation, which are common to image generative models.

To address these challenges, we propose MolFLAE (Molecule Fixed Length AutoEncoder), a Variational AutoEncoder (VAE) for 3D molecules that learns a fixed-dimensional, E(3)-equivariant latent space, independent of atom counts. Our encoder employs an E(3)-equivariant neural network that updates a fixed number of virtual nodes initialized with learnable embeddings, transforming them into fixed-length latent codes for 3D molecules. The latent space is regularized under the standard VAE framework, and a Bayesian Flow Network (BFN) serves as the decoder, reconstructing full molecular structures conditioned on latent codes.

Our autoencoder framework supports a wide range of downstream applications. We first demonstrate that our model can unconditionally generate diverse, valid molecules, achieving competitive performance on standard 3D molecular generation benchmarks. More importantly, our fixed-dimensional latent space enables rich, semantically meaningful manipulations. We show that molecular analogs can be created with varying atom counts, covering simple substitutions to ring contractions. Molecules can also be reconstructed on the shape and orientation of other molecules, yielding chemically plausible outputs. Furthermore, interpolating between two latent codes produces chimera molecules that combine substructures and properties from both parents. Finally, we demonstrate a real-world application of our method in a drug optimization task targeting the human glucocorticoid receptor. We design new molecules that preserve the key binding interactions of known actives while achieving a better balance of potency and hydrophilicity. These results illustrate the flexibility, robustness, and practical utility of our model, highlighting the promise of latent space manipulation as a powerful tool for molecular editing and optimization. Our main contributions are:

- We propose a VAE model that learns a **fixed-dimensional, E(3)-equivariant latent space** for 3D molecular structures;
- The learned latent space enables a **wide range of molecule manipulation tasks**, including analog design, molecule reconstruction, and structure-properties co-interpolation;
- We **introduce quantitative metrics** to evaluate the disentanglement of spatial and semantic latent components, as well as the quality of structural and property interpolation.

## 2 Related Works

**Unconditional Generation for 3D Molecules**    Unconditional 3D molecule generation has achieved rapid advancements, driven by progress in deep generative models. Early works explored autoregressive models to construct molecules atom-by-atom [20, 21]. More recently, inspired by diffusion-like models [22, 23, 24, 25, 26], models like EDM [15], EquiFM [16], and GeoBFN [17], have significantly improved generation quality. VAEs [27] offer an alternative by decoding latent embeddings into molecules, but modeling 3D molecules is challenging due to variable atom counts and the need

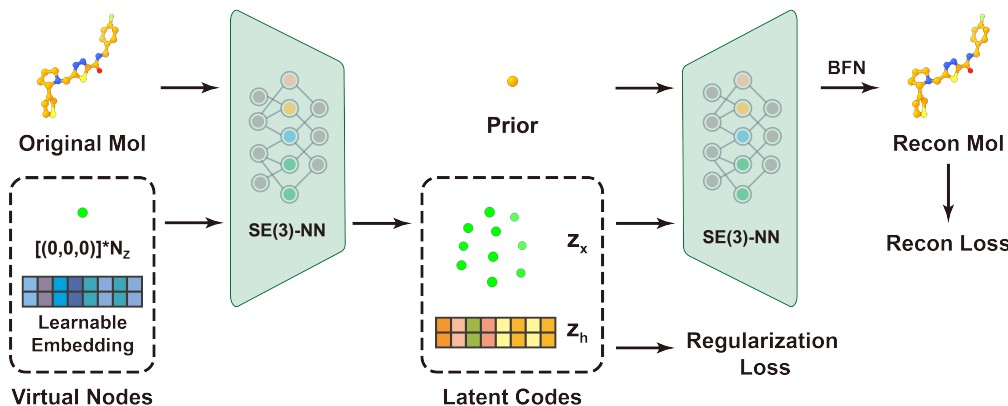

Figure 1: The architecture of MolFLAE. 3D molecules are transformed into latent codes and decoded with BFN. MolFLAE is trained with the recon. loss and the regularization loss of the latent code.

for equivariance of coordinates to rotations and translations. Existing 3D molecular VAEs such as PepGLAD [19] and UniMoMo [28] have made progress by encoding molecular blocks (e.g., amino acid residues) into independent latent nodes. This design ensures E(3)-equivariance but has two key limitations: (1) the number of latent nodes depends on the molecular composition, complicating cross-sample comparisons and interpolation tasks; and (2) spatial relationships between molecular blocks are tightly preserved, restricting the flexibility of the latent space for generative modeling, especially when latent codes are repurposed for tasks like autoregressive modeling by finetuning LLMs.

A key limitation of most unconditional 3D generative models is that their latent spaces vary in length. This variability prohibits operations such as interpolation. Consequently, zero-shot molecule editing becomes non-trivial or even infeasible. To address this, researchers have designed specific models for molecule editing or constructed fixed-dimensional latent spaces, which will be discussed in the following sections.

**3D Molecule Editing** 3D molecule editing has traditionally been divided into several specialized subtasks. Linker design focuses on connecting fragments to form valid new compounds [6, 7], while scaffold inpainting involves masking molecule cores and regenerating them [8]. These two tasks share similarities and can be unified under a mask-prediction framework. However, other editing tasks are harder to generalize. For example, DecompOpt [29] decomposes ligands into substructures and trains a diffusion model conditioned on these substructures, enabling deconstruction-reconstruction of 3D molecules. Another task is property-guided molecule optimization, where molecules are directly edited to improve specific properties using explicit guidance signals. Approaches such as gradient-based optimization [9] and classifier-free guidance [10] have been explored to this end.

While these task-specific methods have demonstrated strong performance, they often lack flexibility and generality. This raises the question if we can achieve more elegant and general 3D molecule manipulation by navigating in the latent space. This motivation has led researchers to explore unified, fixed-length, and semantically meaningful latent spaces for 3D molecules.

**Generate 3D Molecules from Fixed-Dimensional Latent Spaces** Fixed-length autoencoders for molecules have been explored through voxel-based models, which discretize 3D space into uniform grids and apply 3D CNNs [30, 31] or neural fields [32]. While straightforward, these methods are not E(3)-equivariant and struggle to disentangle semantic features from orientations, making molecule manipulation in the latent space not flexible. Local-frame-based models [33] represent conformations with SE(3) invariant features, including distances, bond angles, and dihedrals, making outputs orientation-agnostic. UAE-3D [34] has proposed a 3D fixed-length latent space by discarding the inductive bias of geometric equivariance, yet their latent space suffers from similar feature entanglement problem as voxel-based models. A recent work [35] also constructs fixed-length autoencoder, but applies global pooling over atom features, discarding spatial information that is critical for reconstruction and interaction modeling.

In contrast, our method preserves equivariance without requiring data augmentation, contains spatial information, avoids voxelization, and partially disentangles spatial and semantic information which enables unconditional generation and zero-shot molecule editing.

## 3 Methodology

We encode a variable-size 3D molecule by concatenating it with a set of learnable virtual nodes and updating them together via an E(3)-equivariant neural network to obtain fixed-length embeddings. The regularization loss $\mathcal{L}_{\text{reg}}$ comes from an Multi-Layer Perceptron (MLP) predicting the means and variances to form a variational posterior, from which we sample latent codes that condition a Bayesian Flow Network (BFN) decoder for reconstruction producing the reconstruction loss $\mathcal{L}_{\text{recon}}$. The MolFLAE is trained end-to-end by minimizing

$$\mathcal{L} = \mathcal{L}_{\text{recon}} + \mathcal{L}_{\text{reg}}. \tag{1}$$

We recall the classical Variational AutoEncoder [27] (VAE) in Sec. 3.1 that inspires the above loss. Then introduce BFN in Sec. 3.2. And we introduce the encoder and decoder of MolFLAE in Sec. 3.3 Sec. 3.4.

### 3.1 Variational AutoEncoder

Let $q_\theta(\mathbf{z} \mid \mathbf{x})$ denote the encoder, $p_\phi(\mathbf{x} \mid \mathbf{z})$ the decoder, and $p(\mathbf{z})$ the prior distribution of the latent codes, which is typically chosen as a standard Gaussian. The VAE loss is the negative Evidence Lower Bound (ELBO):

$$\mathcal{L}_{\text{VAE}}(\theta, \phi; \mathbf{x}) = \underbrace{\mathbb{E}_{q_\theta(\mathbf{z}|\mathbf{x})} \left[ -\log p_\phi(\mathbf{x} \mid \mathbf{z}) \right]}_{\text{reconstruction loss}} + \underbrace{D_{\text{KL}} \left( q_\theta(\mathbf{z} \mid \mathbf{x}) \,\|\, p(\mathbf{z}) \right)}_{\text{regularization loss}} \tag{2}$$

Our training loss Eq. 1 is inspired by the above VAE loss.

### 3.2 Bayesian Flow Network

The Bayes Flow Network (BFN) [26] incorporates Bayesian inference to modify the parameters of a collection of independent distributions and using a neural network to integrate the contextual information. Unlike standard diffusion models [22, 23] that primarily handle continuous data via Gaussian noise, BFN extends this paradigm to support both continuous and discrete data types, including categorical variables. This flexibility makes BFN particularly suitable for modeling 3D molecules [17], where the data naturally consists of mixed modalities of continuous coordinates and discrete atom types.

Unlike traditional generative models that operate directly on data, BFN performs inference in the space of distribution parameters. Given a molecule $\mathbf{m}$, the *sender distribution* $p_S(\mathbf{y} \mid \mathbf{m}; \alpha)$ transforms it into a parameterized noisy distribution by adding noise analogous to the forward process in diffusion models. A brief introduction of BFN can be found in Appendix C.1.

### 3.3 MolFLAE Encoder

Inspired by the autoencoder in natural languages models [36] who uses of [CLS] tokens for context compression, we append $N_Z$ learnable virtual nodes to the molecule's point cloud and treat the concatenation of their final embeddings as our fixed-length latent codes.

We denote the coordinates and atom type feature of the $i$-th atom in molecule $\mathcal{M}$ by $\mathbf{x}_M^{(i)} \in \mathbb{R}^3$ and $\mathbf{v}_M^{(i)} \in \mathbb{R}^{D_M}$, respectively. The full molecular input and the learnable virtual nodes are represented as $\mathcal{M} = [\mathbf{x}_M, \mathbf{v}_M]$ and $\mathcal{Z} = [\mathbf{x}_Z, \mathbf{v}_Z]$. The virtual nodes are some artificial atoms with the same size as the real atoms.

We remark that, in order to effectively encode the 3D configuration and chirality of a molecule, the number of virtual nodes $N_Z$ must be at least 4 so that they can form a non-degenerate simplex in 3D space. To ensure sufficient capacity for capturing complex spatial structures, we set $N_Z = 10$.

We employ an E(3)-equivariant neural network $\phi_\theta$ to jointly encode the original molecular point cloud and the appended virtual nodes $(\mathcal{M}, \mathcal{Z})$. After rounds of update, we discard the embeddings of $\mathcal{M}$ and retain only those of $\mathcal{Z}$ as our fixed-length latent representation. In explicit,

$$( \_, [\mathbf{z}_x, \mathbf{z}_h]) = \phi_\theta ([\mathbf{x}_M, \mathbf{v}_M], [\mathbf{x}_Z, \mathbf{v}_Z]) . \tag{3}$$

The latent code is denoted as $[\mathbf{z}_x, \mathbf{z}_h] \in \mathbb{R}^{N_Z \times (3+D_f)}$, where $\mathbf{z}_x$ and $\mathbf{z}_h$ represent the spatial and feature components, $D_f$ is the embedding dimension of features. Note that we only keep the fixed-length part of the $\phi_\theta$ output. So we obtain a fixed-length encoding of the molecules. The network structure and E(3)-equivariance discussion can be found in the Appendix A.

To regularize the latent space, we adopt a VAE formulation. While the initial output $[\mathbf{z}_x, \mathbf{z}_h]$ is deterministic, this can lead to irregular latent geometry and poor interpolatability. To address this, we predict a coordinate-wise Gaussian distribution for each latent dimension:

$$\boldsymbol{\mu}_x = \mathbf{z}_x, \quad [\boldsymbol{\sigma}_x^2, \boldsymbol{\mu}_h, \boldsymbol{\sigma}_h^2] = \texttt{Linear}(\mathbf{z}_h). \tag{4}$$

The resulting latent posterior is regularized via a KL divergence to a fixed spherical Gaussian prior $\mathcal{N}([0, 0], [\text{var}_x, \text{var}_h] I)$, giving rise to the regularization loss:

$$\mathcal{L}_{\text{reg}} = \text{KL} \left( \mathcal{N}([\boldsymbol{\mu}_x, \boldsymbol{\mu}_h], [\boldsymbol{\sigma}_x^2, \boldsymbol{\sigma}_h^2]) \,\big\|\, \mathcal{N}([0, 0], [\text{var}_x, \text{var}_h] \boldsymbol{I}) \right) , \tag{5}$$

where $\text{var}_x, \text{var}_h$ are two fixed scale parameters. This regularization encourages the latent space to be smooth and continuous, facilitating interpolation between molecules and improving the robustness and diversity of samples generated from the prior distribution. In practice, we project (using the linear layer) the feature embedding $\mathbf{z}_h \in \mathbb{R}^{N_Z \times D_f}$ to $\boldsymbol{\mu}_h \in \mathbb{R}^{N_Z \times D_z}$. For notational simplicity, we continue to denote the sampled latent code from $\mathcal{N}(\boldsymbol{\mu}_h, \boldsymbol{\sigma}_h^2 \boldsymbol{I})$ as $\mathbf{z}_h$. The full expression of the regularization loss is provided in Appendix B.

### 3.4 MolFLAE Decoder

The encoder defines a Gaussian posterior over the latent space. We sample a latent code $(\mathbf{z}_x, \mathbf{z}_h)$ from this distribution and use it as the conditioning input to the BFN decoder. By comparing the coordinates and atom type of reconstructed molecule with the original input, we compute the reconstruction loss:

$$\mathcal{L}_{\text{recon}} = \mathcal{L}_x^n + \mathcal{L}_v^n. \tag{6}$$

In our molecular BFN setup, we must jointly model both continuous and discrete aspects of atomic data. This requires a unified representation that enables neural networks to propagate information across modalities while maintaining compatibility with Bayesian updates. It is enough to define a suitable sender distribution $p_S$ allowing the additivity of precision [26].

We model continuous atomic coordinates using Gaussian distributions. Given ground-truth coordinates $\mathbf{x}_M$ and a noise level $\alpha = \rho^{-1}$, the sender generates a noisy observation by adding isotropic Gaussian noise:

$$p_S(\mathbf{y}^x \mid \mathbf{x}_M; \alpha) = \mathcal{N}(\mathbf{y}^x \mid \mathbf{x}_M, \alpha^{-1}\mathbf{I}). \tag{7}$$

For discrete atom types, we model each atom using a categorical distribution over $K$ classes. This distribution is parameterized by a continuous matrix $\boldsymbol{\theta}^v \in \mathbb{R}^{N_M \times K}$, which is transformed into probabilities via a softmax function. Given the ground-truth atom type matrix $\mathbf{e}_{\mathbf{v}_M} = [\mathbf{v}_M^{(1)}, \ldots, \mathbf{v}_M^{(N_M)}]^T \in \mathbb{R}^{N_M \times K}$, where each $\mathbf{v}_M^{(j)}$ is the column one-hot vector representing one of the $K$ atom categories, the sender perturbs it with an artificial Gaussian noise scaled by $\alpha'$, producing:

$$p_S(\mathbf{y}^v \mid \mathbf{v}_M; \alpha') = \mathcal{N}(\mathbf{y}^v \mid \alpha'(K\mathbf{e}_{\mathbf{v}_M} - \mathbf{1}), \alpha' K \boldsymbol{I}) \tag{8}$$

For the initial prior $\boldsymbol{\theta}_0$, we follow [26] and adopt standard Gaussian priors for continuous variables and uniform distributions for categorical ones.

Then we can derive the two loss $\mathcal{L}_x^n + \mathcal{L}_v^n$ respectively [37]. The total forward pass can be found in the Algorithm. 2. The inference process is parallel with the general BFN inference but taking two data modalities into consideration. See Appendix C.4

## 4 Experiments

We train and evaluate MolFLAE on three datasets: QM9 [38], GEOM-Drugs [39] and ZINC-9M (the in-stock subset of ZINC [40] with 9.3M molecules). QM9 contains 134k small molecules with up to 9 heavy atoms, and GEOM-Drugs is a larger-scale dataset featuring 430k drug-like molecules. On both QM9 and GEOM-Drugs experiment, hydrogens are treated explicitly. We use QM9 and GEOM-Drugs to evaluate MolFLAE in unconditional 3D molecule generation task, and demonstrate other applications on the more comprehensive large-scale dataset ZINC-9M,where hydrogens are treated implicitly.

**Unconditional Molecule Generation**  To assess the capability of MolFLAE generate stable, diverse molecules, we first focus on 3D molecule generation task following the setting of prior works [15, 16, 17]. We conduct 10,000 random samplings in the latent space, then decode them into molecules using MolFLAE decoder, subsequently evaluating qualities of these molecules. We sample the atom number from the prior of the training set as previous works like [15]. Table 1 illustrates the benchmark results of unconditional generation with MolFLAE. We also provide results on drug-likeness metrics on GEOM-Drugs in Appendix E, confirming MolFLAE's outstanding performance in generating structurally reasonable and drug-like molecules compared to previous methods.

Table 1: Performance comparison of different methods on the QM9 and GEOM-Drugs dataset.

| # Metrics | QM9 | | | | | GEOM-Drugs | |
|---|---|---|---|---|---|---|---|
| | Atom Sta (%) | Mol Sta (%) | Valid (%) | V×U (%) | Novelty (%) | Atom Sta (%) | Valid (%) |
| Data | 99.0 | 95.2 | 97.7 | 97.7 | - | 86.5 | 99.9 |
| ENF [41] | 85.0 | 4.9 | 40.2 | 39.4 | - | - | - |
| G-Schnet [42] | 95.7 | 68.1 | 85.5 | 80.3 | - | - | - |
| GDM-AUG [15] | 97.6 | 71.6 | 90.4 | 89.5 | 74.6 | 77.7 | 91.8 |
| EDM [15] | 98.7 | 82.0 | 91.9 | 90.7 | 58.0 | 81.3 | 92.6 |
| EDM-Bridge [43] | 98.8 | 84.6 | 92.0 | 90.7 | - | 82.4 | 92.8 |
| GEOLDM [18] | 98.9 | 89.4 | 93.8 | 92.7 | 57.0 | 84.4 | 99.3 |
| GEOBFN $_{50}$ [17] | 98.3 | 85.1 | 92.3 | 90.7 | 72.9 | 75.1 | 91.7 |
| GEOBFN $_{100}$ [17] | 98.6 | 87.2 | 93.0 | 91.5 | 70.3 | 78.9 | 93.1 |
| UniGEM [44] | 99.0 | 89.8 | 95.0 | **93.2** | - | 85.1 | 98.4 |
| MolFLAE $_{50}$ | 99.3 | 90.4 | 95.9 | 92.1 | **77.1** | **86.9** | 99.2 |
| MolFLAE $_{100}$ | **99.4** | **92.0** | **96.8** | 88.9 | 74.5 | 86.7 | **99.7** |

Compared with several baseline models, MolFLAE achieves competitive performance across atom stability, molecular stability, and validity metrics on both QM9 and GEOM-Drugs dataset, while requiring fewer sampling steps. These results suggest that our latent space is well-structured, supporting efficient and reliable molecular generation.

**Generating Analogs with Different Atom Numbers**  First, we probe the smoothness of MolFLAE decoder to atom numbers by forcing the generation with increased or decreased atom numbers based on the original latent code. We examine the similarities between generated molecules and original molecules with MCS-IoU (Maximum Common Substructure Intersection-over-Union). Generated molecules share similar orientations, shapes and 2D structures with the original input, validating the desired smoothness. Detailed results are presented in Table 2, and three examples are provided in Fig. 2 for better illustration.

Table 2: Evaluating 2D similarities between generated analogs and original molecules.

| Atom Number | -2 | -1 | 0 | 1 | 2 |
|---|---|---|---|---|---|
| MCS-IoU similarity | 69.79 | 76.69 | 84.08 | 76.05 | 69.95 |
| Valid(%) | 100.0 | 99.89 | 99.76 | 99.89 | 99.68 |
| Atom Sta(%) | 84.58 | 83.28 | 82.48 | 82.38 | 82.53 |

**Exploring the disentanglement of the latent space via molecule reconstruction**  The latent space of MolFLAE consists of two parts, the E(3)-equivariant component $\mathbf{z}_x$ and the E(3)-invariant component $\mathbf{z}_h$. Ideally, spatial and semantic features of molecules disentangle spontaneously, with $\mathbf{z}_x$ encoding the shape and orientation and $\mathbf{z}_h$ encodes substructures of molecules.

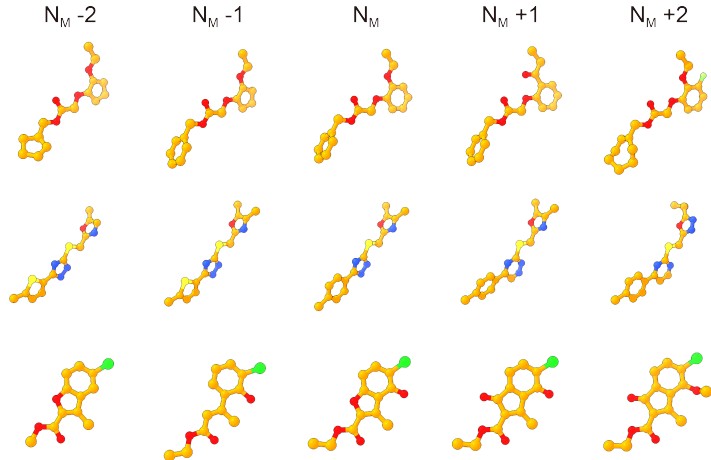

Figure 2: Examples for analog generation with variable atom numbers.

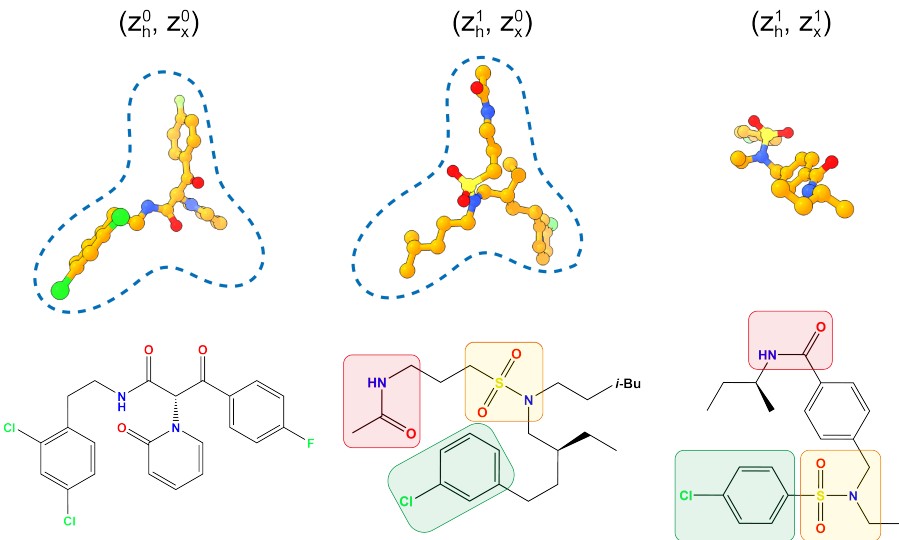

Figure 3: An example for molecule reconstruction with new shape and orientation.

In this section, we explore this disentanglement hypothesis by swapping $\mathbf{z}_x$ and $\mathbf{z}_h$ between molecules and observe decoded molecules. Formally, with two molecules $M_0$ and $M_1$, and their latent $\left(\mathbf{z}_h^0, \mathbf{z}_x^0\right)$ and $\left(\mathbf{z}_h^1, \mathbf{z}_x^1\right)$, we decode molecules with $\left(\mathbf{z}_h^0, \mathbf{z}_x^1\right)$ (named preserving $\mathbf{z}_h$) and $\left(\mathbf{z}_h^1, \mathbf{z}_x^0\right)$ (named preserving $\mathbf{z}_x$), respectively.

In experiments, we have observed a partial disentanglement of the MolFLAE latent space. As shown in Fig. 3, following the disentanglement hypothesis, substructures information of $M_1$ should be able to be extracted by isolating $\mathbf{z}_h^1$. As $\mathbf{z}_h^1$ is not sufficient to reconstruct $M_1$, we view it as a deconstructed molecule. Then, we reconstruct these substructures into the shape and orientation of $M_0$, by decoding the hybrid latent code $\left(\mathbf{z}_h^1, \mathbf{z}_x^0\right)$. The resulted molecule shares a similar shape and orientation with $M_0$, indicated by the dash lines in Fig 3. Moreover, it also shares similar substructures as $M_1$ like amide, chlorobenzyl, sulfamide, indicated by rectangles of corresponding colors.

Quantitatively, we compute the MACCS fingerprint [45] similarity (considering substructure overlapping) and *in situ* shape similarity (considering shape, orientation and relative position) of 1000 hybrid molecules $\left(\mathbf{z}_h^1, \mathbf{z}_x^0\right)$ and $\left(\mathbf{z}_h^0, \mathbf{z}_x^1\right)$ with the original molecule $\left(\mathbf{z}_h^0, \mathbf{z}_x^0\right)$ (Table 3). Under the setting of preserving $\mathbf{z}_x$, the shape similarity is significantly higher than the preserving $\mathbf{z}_h$ (0.394 vs 0.174); indicating $\mathbf{z}_x$ indeed encodes shape and orientation information. Similarly, MACCS similarity is

higher under the preserving $\mathbf{z}_h$ setting than the preserving $\mathbf{z}_x$ setting (0.580 vs 0.421). These results support that the latent space of MolFLAE is partially disentangled, with $\mathbf{z}_h$ representing substructure composition and $\mathbf{z}_x$ representing shape and orientation.

Table 3: Measuring molecule reconstruction similarities under different settings.

| Metrics | Preserving $\mathbf{z}_x$ | | | | Preserving $\mathbf{z}_h$ | | | |
|---|---|---|---|---|---|---|---|---|
| | MACCS Sim ↓ | Shape Sim ↑ | Valid(%)(↑) | Atom Sta(%)(↑) | MACCS Sim ↑ | Shape Sim ↓ | Valid(%)(↑) | Atom Sta(%)(↑) |
| MolFLAE | 0.421 | 0.394 | 100.0 | 85.20 | 0.580 | 0.174 | 100.0 | 84.62 |

**Latent Interpolation**  The fixed-dimensional latent space allows flexible manipulation of molecular representations via vector convex combinations. In MolFLAE, the regularization loss further encourages smooth transitions between latent codes, facilitating continuous transformations between molecules. Fig. 4 presents the interpolation of three pairs of molecules, indicating smooth transformations of shape and orientation.

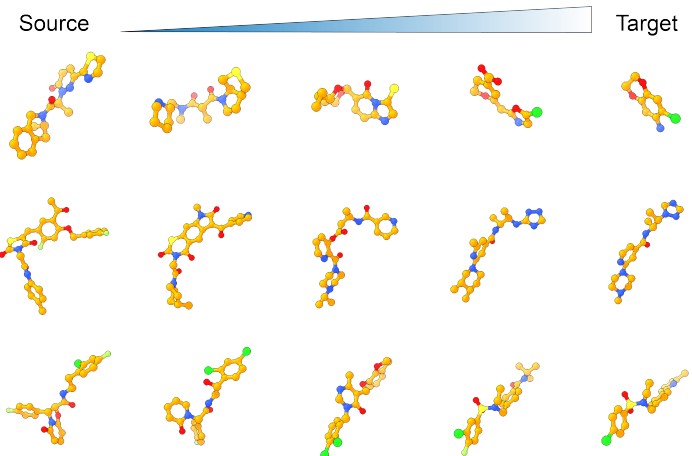

Figure 4: Examples for the latent interpolation between molecules.

To further quantify the quality of interpolation, We evaluate the trend of properties of the intermediate molecules during the transformation, considering both structural and physical properties (detailed descriptions are provided in Appendix D). We compute the Pearson correlation coefficient $r$ between each property value and its corresponding interpolation index (adjusted with the sign of property difference between the source and target molecule). We also report the associated $p$-values from the Pearson significance test to assess the statistical evidence for linear trends. A property is considered to exhibit significant linear variation along the interpolation trajectory if the null hypothesis of zero correlation is rejected at the 5% significance level, i.e., if $-\log p > -\log(0.05) \approx 1.3$. Detailed results are documented in Table 5 and Table 4. We also report the step-wise molecular validity and atom stability during interpolation in Table 6, which indicates that most intermediate molecules are valid and stable.

Table 4: Monitoring the trend of structural properties along with molecule interpolations.

| Interpo Num | Similarity Preference | | sp3frac | | BertzCT | | QED | |
|---|---|---|---|---|---|---|---|---|
| | Pearson's r | -log$p$ | Pearson's r | -log$p$ | Pearson's r | -log$p$ | Pearson's r | -log$p$ |
| 8 | 0.9261 | 3.3346 | 0.4314 | 0.8518 | 0.6537 | 1.7888 | 0.5460 | 1.2128 |
| 10 | 0.9191 | 4.1340 | 0.3982 | 0.9639 | 0.6344 | 2.1138 | 0.5194 | 1.4228 |
| 12 | 0.9122 | 4.8476 | 0.3809 | 1.0889 | 0.6281 | 2.4728 | 0.5059 | 1.6216 |

Table 5: Monitoring the trend of physical properties along with molecule interpolations.

| Interpo Num | Labute ASA | | TPSA | | LogP | | MR | |
|---|---|---|---|---|---|---|---|---|
| | Pearson's r | -log$p$ | Pearson's r | -log$p$ | Pearson's r | -log$p$ | Pearson's r | -log$p$ |
| 8 | 0.9067 | 4.6366 | 0.5711 | 1.3783 | 0.5400 | 1.2363 | 0.8467 | 3.5188 |
| 10 | 0.9041 | 5.7687 | 0.5620 | 1.6533 | 0.5216 | 1.4533 | 0.8388 | 4.3350 |
| 12 | 0.8939 | 6.8567 | 0.5425 | 1.8572 | 0.4925 | 1.6259 | 0.8255 | 5.1216 |

Table 6: Step-wise Validity and Atom Stability during interpolation.

| Interpo Num | Metrics | Step | | | | | | | | | | | |
|---|---|---|---|---|---|---|---|---|---|---|---|---|---|
| | | 1 | 2 | 3 | 4 | 5 | 6 | 7 | 8 | 9 | 10 | 11 | 12 |
| 8 | Valid(%) | 99.90 | 99.90 | 99.90 | 100.0 | 99.90 | 100.0 | 100.0 | 100.0 | | | | |
| | Atom Sta(%) | 82.23 | 82.95 | 84.17 | 84.57 | 84.21 | 83.74 | 83.10 | 82.63 | | | | |
| 10 | Valid(%) | 100.0 | 99.90 | 99.90 | 99.80 | 99.80 | 100.0 | 100.0 | 100.0 | 99.80 | 99.90 | | |
| | Atom Sta(%) | 82.23 | 82.75 | 83.53 | 84.07 | 84.36 | 84.35 | 84.16 | 84.17 | 82.78 | 82.67 | | |
| 12 | Valid(%) | 99.90 | 100.0 | 99.90 | 100.0 | 100.0 | 100.0 | 100.0 | 100.0 | 99.90 | 99.90 | 100.0 | 99.80 |
| | Atom Sta(%) | 82.39 | 82.50 | 83.76 | 83.67 | 83.92 | 84.65 | 84.20 | 84.50 | 84.27 | 83.00 | 82.84 | 82.57 |

**Applying MolFLAE to optimize molecules targeting the human glucocorticoid receptor (hGR)**
To assess the real-world utility of our method, we applied our method to optimize drug candidates for the hGR, which requires balancing hydrophobic-interaction-centric binding with aqueous solubility. The hGR is a key target for anti-inflammatory, and our optimization starts from two known actives. AZD2906 is a potent hGR modulator but is poorly soluble [46], while BI-653048 is more soluble but less potent [47]. To showcase the performance of our model, We computationally evaluate the potency and hydrophilicity with Glide docking and QikProp CLogPo/w from the Schrodinger Suite, where lower docking scores indicate better potency, and lower CLogPo/w values indicate better hydrophilicity. These computational metrics align with real-world properties of AZD2906 and BI-653048. AZD2906 has a docking score of -13.16 and a high CLogPo/w of 5.61, whereas BI-653048 shows a better CLogPo/w of 3.90 but a weaker docking score of -10.62 (Fig. 5B and C). These results facilitate the computational evaluation of MolFLAE generated molecules.

To explore trade-offs, we blended these two molecules in latent space using 90% AZD2906 and 10% BI-653048, generating 100 candidates. The top 10 molecules outperformed BI-653048 in docking score, and 8 also improved hydrophilicity (CLogPo/w) compared to AZD2906. These candidates preserved AZD2906's binding shape while introducing polar groups for better solubility. Sample 34, for instance, it retained key pharmacophores of AZD2906 and BI-653048 for interacting with the receptor (indicated by colored rectangles for each pharmacophore), achieving a balanced property with its docking score of -11.15 and CLogPo/w of 3.75 (Fig. 5F). Moreover, its docking pose closely matched both its generated conformation (RMSD = 1.35 Å) and AZD2906's crystal structure (Fig. 5D and E), representing the advantage of explicitly modeling of 3D coordinates by MolFLAE. These results highlight our method's potential for meaningful molecular optimization and drug design.

## 5 Conclusion and Future Works

In this work, we present MolFLAE, a flexible VAE framework for manipulating 3D molecules within a fixed-dimensional, E(3)-equivariant latent space. Our method demonstrates strong performance across multiple tasks, including unconditional generation, analog design, substructure reconstruction, and latent interpolation. We further validate the real-world utility of MolFLAE through a case study on generating drug-like molecules targeting the hGR, balancing potency and solubility.

Beyond the reported experiments, MolFLAE naturally extends to a wider range of tasks. For example, molecule inpainting can be achieved by encoding discontinuous fragments and decoding to larger atom sets. Structural superposition can be achieved efficiently via the weighted Kabsch algorithm on latent nodes, avoiding the high complexity of atom-wise bipartite matching. These applications are exemplified in Fig 6, and a deeper exploration is left for future work due to space constraints.

While MolFLAE demonstrates strong performance across multiple tasks, there remains room for improvement in the disentanglement and interpretability of its latent space. We hypothesize that better disentanglement can be achieved by enforcing the invariance of semantic latent $\mathbf{z}_h$ to molecular

conformational changes or other non-rigid perturbations. Future works may explore incorporating self-contrastive objectives to better capture chemical semantics with latent representations.

In summary, our results highlight the versatility of MolFLAE and its promise as a general-purpose framework for 3D molecule generation and editing. This work opens new directions for exploring the broader applications of fixed-dimensional, E(3)-equivariant latent spaces in molecular modeling.

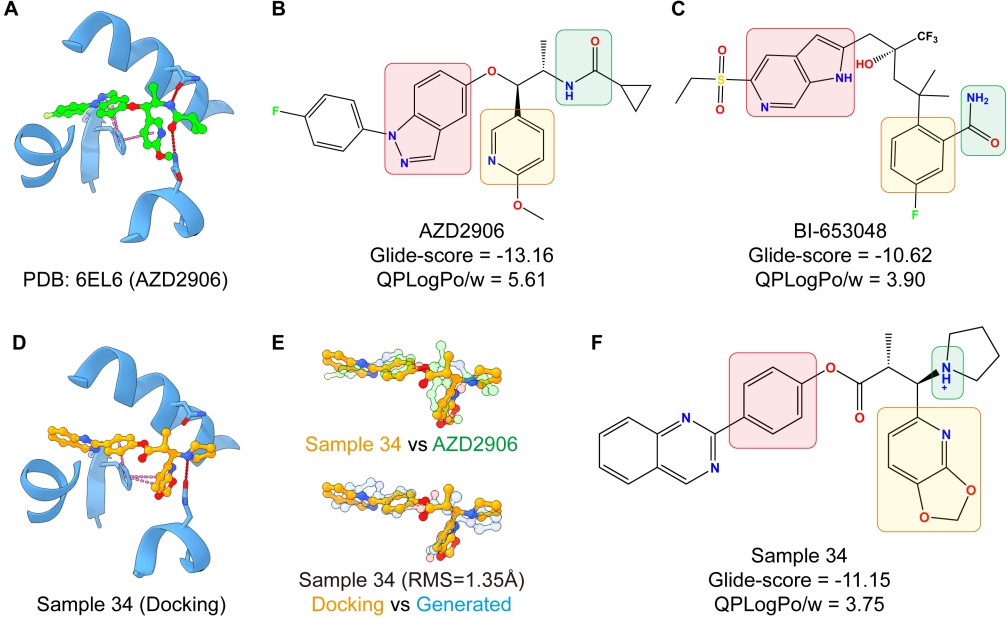

Figure 5: Applying MolFLEA to optimizing AZD2096 targeting the hGR. **A,** the cystal structure of AZD2096 in complex with hGR. **B, C, and F,** 2D structures of AZD2096, BI-653048, and sample 34, with their docking score and CLogPo/w. **D,** the docking pose of sample 34. **E,** comparing the docking pose of sample 34 with AZD2096 and its generated pose before docking.

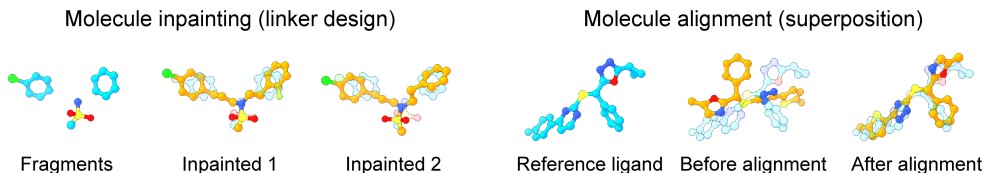

Figure 6: Exemplifying the application of MolFLAE to molecule inpainting and superposition.

## Acknowledgements

This work is supported by Beijing Academy of Artificial Intelligence and Beijing Frontier Research Center for Biological Structure Fundings.

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

# A   Neural Network Details

We adopt the network structure of MolCRAFT [37] as our E(3)-Equivariant graph neural network (GNN) backbone. Originally designed to model interactions between ligand and protein pocket atoms, the network distinguishes between *update nodes* $\mathbf{u}$ (whose coordinates are updated) and *condition nodes* $\mathbf{c}$ (which provide contextual information and the coordinates are not updated).

## A.1   Neural Network Architecture

To align with our autoencoder framework, we adapt this formulation by assigning roles to nodes based on the encoding or decoding stage. In the encoder, the update nodes correspond to the virtual nodes, while the condition nodes are the atoms of the input ground truth molecule. Conversely, in the decoder, the update nodes are the molecular atoms being generated, and the condition nodes are the latent codes produced by the encoder.

To be convenient, we concatenate the spatial part and feature part of the update nodes $[\mathbf{x}_{\text{update}}, \mathbf{v}_{\text{update}}]$ and condition nodes $[\mathbf{x}_{\text{condition}}, \mathbf{v}_{\text{condition}}]$ with writing $\mathbf{x}^{\ell} = [\mathbf{x}^{\ell}_{\text{update}}, \mathbf{x}^{\ell}_{\text{condition}}]$ and $\mathbf{h}^{\ell} = [\mathbf{v}^{\ell}_{\text{update}}, \mathbf{v}^{\ell}_{\text{condition}}]$, where the superscript represents the $\ell$-th layer of $\phi$, $0 \leq \ell \leq L$. The Initial hidden embedding $\mathbf{h}^{0}$ is obtained by an MLP embedding layer that encodes the atom feature $[\mathbf{h}]$. No embedding layer for the atom spatial coordinates. The construction of $\phi_{\theta}$ is alternately updating the atom feature embeddings $\mathbf{h}$ and coordinates $\mathbf{x}$ as

$$\mathbf{h}_i^{\ell+1} = \mathbf{h}_i^{\ell} + \sum_{\substack{j \in \mathcal{V}(i) \\ j \neq i}} f_h\big(d_{ij}^{\ell}, \, \mathbf{h}_i^{\ell}, \, \mathbf{h}_j^{\ell}, \, \mathbf{e}_{ij}; \theta_h\big), \tag{9}$$

$$\mathbf{x}_i^{\ell+1} = \mathbf{x}_i^{\ell} + \sum_{\substack{j \in \mathcal{V}(i) \\ j \neq i}} \big(\mathbf{x}_i^{\ell} - \mathbf{x}_j^{\ell}\big) f_x\big(d_{ij}^{\ell}, \, \mathbf{h}_i^{\ell+1}, \, \mathbf{h}_j^{\ell+1}, \, \mathbf{e}_{ij}; \theta_x\big) \mathbf{1}_{\text{update}}. \tag{10}$$

Here, $\mathcal{V}(i)$ is the neighbors of $i$, who could have information communication to $i$. We choose the $k$ nearest nodes from $i$. $d_{ij}^{\ell} = \|\mathbf{x}_i^{\ell} - \mathbf{x}_j^{\ell}\|_2$ denotes the Euclidean distance between atoms $i$ and $j$ at layer $\ell$, and $\mathbf{e}_{ij}$ encodes whether the pair $(i, j)$ belongs to the update nodes, condition nodes, or the connection between them. The indicator $\mathbf{1}_{\text{update}}$ ensures that coordinate updates are only applied to update nodes, keeping the positions of condition nodes fixed.

## A.2   E(3)-Equivariance Discussion

In molecular modeling, it is essential that the learned distribution over update nodes be invariant to translations, reflections, and rotations of the condition nodes. This E(3)-invariance reflects a fundamental inductive bias in molecular systems [15, 48]. Since SE(3) is a subgroup of E(3), any E(3)-equivariant model is also SE(3)-equivariant. While some prior works (e.g., [37, 7]) adopt the term SE(3), others (e.g., [15]) use E(3), which more accurately describes the symmetry of their networks. We follow the latter to avoid ambiguity.

The full Euclidean group in $\mathbb{R}^3$, denoted E(3), consists of rigid-body transformations of the form $T(x) = Rx + t$, where $R \in \mathbb{R}^{3 \times 3}$ is a orthogonal matrix and $t \in \mathbb{R}^3$ is a translation vector.

If we pre-align the condition nodes by centering them at their center of mass (i.e., eliminating the translational degree of freedom), then the resulting likelihood becomes E(3)-invariant under the following condition:

**Proposition A.1** (Proposition 4.1 in [37]). *Let $T \in \mathrm{E}(3)$ denote a rigid transformation. If the condition nodes are centered at zero and the parameterization $\boldsymbol{\Phi}(\boldsymbol{\theta}, \mathbf{c}, t)$ is E(3)-equivariant, then the likelihood is invariant under $T$:*

$$p_{\phi}(T(\mathbf{u}) \mid T(\mathbf{c})) = p_{\phi}(\mathbf{u} \mid \mathbf{c}).$$

This property ensures that the decoder's predictions respect the underlying geometric symmetries of molecular structures, which is crucial for both sample quality and spatial information learning of latent codes.

## A.3 Encoder Network

In the encoder network, the update nodes are the virtual nodes while the condition nodes are the input ground truth molecule.

Before passing into the network, we apply a linear layer to embed the one-hot atom features $\mathbf{v}_M \in \mathbb{R}^{N_M \times D_M}$ into a continuous feature space $\mathbb{R}^{N_M \times D_f}$, where $D_f$ denotes the embedding dimension. The virtual node features are initialized as learnable parameters in the same embedded space $\mathbb{R}^{N_Z \times D_f}$ and are only defined at the embedding level. Their initial spatial positions are set to zero.

After $L$ layers of message passing, the output of the Network $\phi_\theta$ is given by

$$\phi_\theta\left([\mathbf{x}_M, \mathbf{v}_M], [\mathbf{x}_Z, \mathbf{v}_Z]\right) = \left([\tilde{\mathbf{x}}_M, \tilde{\mathbf{v}}_M], [\tilde{\mathbf{x}}_Z, \tilde{\mathbf{v}}_Z]\right) = \left(\_, [\mathbf{z}_x, \mathbf{z}_h]\right), \tag{11}$$

where only the virtual node outputs $[\mathbf{z}_x, \mathbf{z}_h]$ are retained as the final latent code. Since the coordinate updates are designed to be E(3)-equivariant at each layer, the entire encoder $\phi_\theta$ is equivariant by construction.

Importantly, we do not apply a softmax projection to convert feature embeddings into one-hot vectors. Instead, we preserve the continuous representations to retain richer information for downstream generation.

Then we apply a VAE layer to furtherly encode the latent code to a Gaussian distribution. See Appendix B.

## A.4 Decoder Network

The decoder follows a mirrored architecture, where the update nodes correspond to the generated molecule, and the conditioning nodes are the latent virtual codes. Similar to the encoder, we discard the outputs corresponding to the latent nodes and retain only the decoded molecule representations for final output. The atom number $N_M$ is to be known beforehand. We can sample it from the training set prior when doing unconditional generation [15] or edit it in generating analogs with different atom numbers.

# B VAE Details

We adopt the regularization loss to regularize the latent space. Given the deterministic initial output $[\mathbf{z}_x, \mathbf{z}_h]$, we predict a coordinate-wise Gaussian distribution for each latent dimension:

$$\boldsymbol{\mu}_x = \mathbf{z}_x, \quad [\boldsymbol{\sigma}_x^2, \ \boldsymbol{\mu}_h, \boldsymbol{\sigma}_h^2] = \texttt{Linear}(\mathbf{z}_h). \tag{12}$$

where we assume isotropic variance for 3D coordinates (i.e., each atom shares a scalar variance across $x, y, z$), while the feature dimensions $\mathbf{z}_h$ are assigned independent variances per entry. This design ensures that the latent distribution preserves equivariance in the spatial domain while maintaining expressiveness in the feature domain.

In practice, we project (using the linear layer) the feature embedding $\mathbf{z}_h \in \mathbb{R}^{N_Z \times D_f}$ to $\boldsymbol{\mu}_h \in \mathbb{R}^{N_Z \times D_Z}$. For notational simplicity, we continue to denote the sampled latent code from $\mathcal{N}(\boldsymbol{\mu}_h, \boldsymbol{\sigma}_h^2 \boldsymbol{I})$ as $\mathbf{z}_h$.

The resulting latent posterior is regularized via a KL divergence to a fixed spherical Gaussian prior $\mathcal{N}([0,0], [\text{var}_x, \text{var}_h]\boldsymbol{I})$, giving rise to the regularization loss:

$$\mathcal{L}_{\text{reg}} = \text{KL}\left(\mathcal{N}([\boldsymbol{\mu}_x, \boldsymbol{\mu}_h], [\boldsymbol{\sigma}_x^2, \boldsymbol{\sigma}_h^2]) \ \middle\| \ \mathcal{N}([0,0], [\text{var}_x, \text{var}_h]\boldsymbol{I})\right), \tag{13}$$

where $\text{var}_x, \text{var}_h$ are two fixed scale parameters. Since the KL-divergence between two Gaussian distribution $P_i \sim \mathcal{N}(\mu_i, \sigma_i^2)$, $i = 1, 2$ is

$$D_{KL}(P_1 \| P_2) = \log \frac{\sigma_2}{\sigma_1} + \frac{\sigma_1^2 + (\mu_1 - \mu_2)^2}{2\sigma_2^2} - \frac{1}{2}. \tag{14}$$

Hence the regularization loss is $\mathcal{L}_{\text{reg}} = \mathcal{L}_{\text{KL}}^{(h)} + \mathcal{L}_{\text{KL}}^{(x)}$, where

$$\mathcal{L}_{\text{KL}}^{(h)} = \sum_{i=1}^{N_Z \times D_Z} \frac{1}{2} \left( \frac{\mu_i^2 + \sigma_i^2}{\text{var}_h} - \log \sigma_i^2 - 1 \right), \tag{15}$$

$$\mathcal{L}_{\text{KL}}^{(x)} = \sum_{i=1}^{N_Z \times 3} \frac{1}{2} \left( \frac{\mu_i^2 + \sigma_i^2}{\text{var}_x} - \log \sigma_i^2 - 1 \right). \tag{16}$$

This regularization encourages the latent space to be smooth and continuous, facilitating interpolation between molecules and improving the robustness and diversity of samples generated from the prior distribution.

## C  BFN details

### C.1  A Brief Introduction to Bayes Flow Network

BFN can be interpreted as a communication protocol between a sender and a receiver. The sender observes the ground-truth molecule $\mathbf{m}$ and deliberately adds noise to obtain a corrupted version $\mathbf{y}$, which is transmitted to the receiver. Given the known precision level (e.g., $\alpha$ from a predefined schedule $\beta(t)$), the receiver performs Bayesian inference and leverages a neural network to incorporate contextual information, producing an estimate of $\mathbf{m}$. The communication cost at each timestep is defined as the KL divergence between the sender's noising distribution and the corrupted output of the receiver's updated belief. By minimizing this divergence across timesteps, the receiver learns to approximate the posterior and generate realistic samples from prior noise using the same learned inference process.

### C.1.1  Training of BFN

Concretely, at each communication step $t_i$, the sender perturbs $\mathbf{m}$ using the *Sender distribution* (adding noise distribution) $p_S(\mathbf{y}_i \mid \mathbf{m}; \alpha_i)$ to produce a noisy latent $\mathbf{y}_i$, analogous to the forward process in diffusion models. The receiver then reconstructs (using Bayes updates and Neural Networks) $\hat{\mathbf{m}}$ via the *Output distribution*:

$$p_O(\hat{\mathbf{m}} \mid \mathbf{y}_i; \theta) = \mathbf{\Phi}(\boldsymbol{\theta}_{i-1}, \mathbf{z}, t_i), \tag{17}$$

where $\mathbf{\Phi}$ is a neural network which is expected to reconstruct the sample $\hat{\mathbf{m}}$ given the Bayes-updated parameters $\boldsymbol{\theta}_{i-1}$, conditioning code $\mathbf{z}$ and time $t_i$. Then re-applies the same noising process to obtain the *receiver distribution* $p_R(\mathbf{y}_i \mid \boldsymbol{\theta}_{i-1}, \mathbf{z}; t_i) = \mathbb{E}_{\hat{\mathbf{m}} \sim p_O} p_S(\mathbf{y}_i \mid \hat{\mathbf{m}}; \alpha_i)$. The training objective is to minimize the KL divergence $\text{KL}(p_S \| p_R)$ at each step, encouraging consistency between sender and receiver.

In practice, we use Gaussian for Continuous data and categorical distribution for discrete data. Therefore Bayesian updates has closed form. The details can be found in the Appendix C.3. *Bayesian update distribution* $p_U$ stems from the Bayesian update function $h$,

$$p_U(\boldsymbol{\theta}_i \mid \boldsymbol{\theta}_{i-1}, \mathbf{m}, \mathbf{z}; \alpha_i) = \mathbb{E}_{\mathbf{y}_i \sim p_S} \, \delta \left( \boldsymbol{\theta}_i - h(\boldsymbol{\theta}_{i-1}, \mathbf{y}_i, \alpha_i) \right), \tag{18}$$

where $\delta(\cdot)$ is Dirac delta distribution. This expectation eliminates the randomness of the sent sample from the sender.

According to the nice additive property of accuracy [26], the best prediction of $\mathbf{m}$ up to time $t_i$ is the *Bayesian flow distribution* $p_F$ which could be obtained by adding all the precision parameters:

$$p_F(\boldsymbol{\theta}_i \mid \mathbf{m}, \mathbf{z}; t_i) = p_U(\boldsymbol{\theta}_i \mid \boldsymbol{\theta}_0, \mathbf{m}, \mathbf{z}; \beta(t_i)), \text{ where } \beta(t_i) = \sum_{j \leq i} \alpha_j. \tag{19}$$

Therefore, the training objective for $n$ steps is to minimize:

$$L^n(\mathbf{m}, \mathbf{z}) = \mathbb{E}_{i \sim U(1,n)} \, \mathbb{E}_{\mathbf{y}_i \sim p_S, \, \boldsymbol{\theta}_{i-1} \sim p_F} \, D_{\text{KL}}(p_S \, \| \, p_R). \tag{20}$$

### C.1.2 Inference of BFN

During inference, the sender is no longer available to provide noisy samples to help the receiver improve its belief. However, as training minimizes $D_{\mathrm{KL}}(p_S \| p_R)$. Thus, we can reuse the same communication mechanism by iteratively applying the learned receiver distribution $p_R$ to generate samples.

Given prior parameters $\boldsymbol{\theta}_0$, accuracies $\alpha_1, \ldots, \alpha_n$ and corresponding times $t_i = i/n$, the $n$-step sampling procedure recursively generates $\boldsymbol{\theta}_1, \ldots, \boldsymbol{\theta}_n$ by sampling $\mathbf{x}'$ from $p_O(\cdot \mid \boldsymbol{\theta}_{i-1}, \mathbf{z}, t_{i-1})$, $\mathbf{y}$ from $\mathbf{y} \sim p_R(\cdot \mid \boldsymbol{\theta}_{i-1}, \mathbf{z}, t_{i-1}, \alpha_i)$, then setting $\boldsymbol{\theta}_i = h(\boldsymbol{\theta}_{i-1}, \mathbf{z}, \mathbf{y})$, and pass the result to the neural network. The final sample is drawn from $p_O(\cdot \mid \boldsymbol{\theta}_n, \mathbf{z}, 1)$.

This recursive procedure enables BFN to generate molecule from a simple prior, guided solely by the learned receiver network and the latent codes. see Algorithm. 1

However, explicitly sampling $\mathbf{x}'$ from $p_O(\cdot \mid \boldsymbol{\theta}_{i-1}, \mathbf{z}, t_{i-1})$, particularly for discrete data could introduce unnecessary noise and impair the stability of the generation process. Instead, following [37, 49], we directly operate in the parameter space, avoiding noise injection from sampling and enabling a more deterministic and efficient inference. See Algorithm. 3

---

**Algorithm 1:** Inference of General Bayes Flow Networks

**Input:** Initial prior parameter $\boldsymbol{\theta}_0$, noise schedule $\{\alpha_i\}_{i=1}^n$, timestep grid $\{t_i = i/n\}_{i=1}^n$, conditioning latent code $\mathbf{z}$
**Output:** Final molecular sample $\mathbf{x}_{\mathrm{final}}$
**for** $i = 1$ **to** $n$ **do**
    Sample $\mathbf{x}' \sim p_O(\cdot \mid \boldsymbol{\theta}_{i-1}, \mathbf{z}, t_{i-1})$
    Sample $\mathbf{y} \sim p_R(\cdot \mid \boldsymbol{\theta}_{i-1}, \mathbf{z}, t_{i-1}, \alpha_i)$
    Update latent state: $\boldsymbol{\theta}_i \leftarrow h(\boldsymbol{\theta}_{i-1}, \mathbf{z}, \mathbf{y})$
    Apply the network: $\boldsymbol{\theta}_i \leftarrow \boldsymbol{\Phi}(\boldsymbol{\theta}_i, \mathbf{z}, t_i)$
Sample final output: $\mathbf{x}_{\mathrm{final}} \sim p_O(\cdot \mid \boldsymbol{\theta}_n, \mathbf{z}, 1)$

---

### C.2 MolFLAE Reconstruction Loss

BFN can be trained by minimizing the KL-divergence between noisy sample distributions. BFN allows training in discrete time and continuous time, and for efficiency we adopt the $n$-step discrete loss.

Given the ground truth molecule $\mathbf{m} = [\mathbf{x}, \mathbf{v}]$ and its latent code $z$, we can have the reconstruction loss

$$\mathcal{L}_{\mathrm{recon}} = \mathcal{L}_x^n + \mathcal{L}_v^n. \tag{21}$$

The above two summands are coordinates loss and atom type loss:

- Since the atom coordinates and the noise are Gaussian, the loss can be written analytically as follows:

$$\mathcal{L}_x^n = D_{\mathrm{KL}}\left(\mathcal{N}(\mathbf{x}, \alpha_i^{-1}\boldsymbol{I}) \,\|\, \mathcal{N}(\hat{\mathbf{x}}(\boldsymbol{\theta}_{i-1}, \mathbf{z}, t), \alpha_i^{-1}\boldsymbol{I})\right)$$
$$= \frac{\alpha_i}{2} \|\mathbf{x} - \hat{\mathbf{x}}(\boldsymbol{\theta}_{i-1}, \mathbf{z}, t)\|^2 \tag{22}$$

- The atom type loss can also be derived by taking KL-divergence between Gaussians [26], assuming $D_M$ is the number of atom types, $N_M$ is the number of the atom :

$$\mathcal{L}_v^n = \ln \mathcal{N}\left(\mathbf{y}^v \,\big|\, \alpha_i(D_M\mathbf{e}_v - \mathbf{1}), \alpha_i D_M\boldsymbol{I}\right)$$
$$- \sum_{d=1}^{N_M} \ln\left(\sum_{k=1}^{D_M} p_O(k \mid \boldsymbol{\theta}; t)\mathcal{N}\left(.^{(d)} \,\big|\, \alpha_i(D_M\mathbf{e}_k - \mathbf{1}), \alpha_i D_M\boldsymbol{I}\right)\right) \tag{23}$$

Together with two training losses, we can summarize the forward pass. In practice, we use reconstruction and regularization loss weights to get the final loss, see Table. 10

---

**Algorithm 2:** Forward Pass of MolFLAE

---

**Input:** Molecule $\mathcal{M} = (\mathbf{x}_M, \mathbf{v}_M)$, Number of Virtual nodes $N_Z$
**Output:** Reconstruction loss $\mathcal{L}_{\text{recon}}$, Regularization $\mathcal{L}_{\text{reg}}$
**Introduce $N_Z$ virtual nodes:**
$\quad \mathcal{Z} = [\mathbf{x}_Z, \mathbf{v}_Z]$
**E(3)-equivariant encoding:**
$\quad \left(\_, [\mathbf{z}_x, \mathbf{z}_h]\right) \leftarrow \phi_\theta\big((\mathcal{M}, \mathcal{Z})\big)$
**VAE parameterisation:**
$\quad \boldsymbol{\mu}_x \leftarrow \mathbf{z}_x, \quad [\boldsymbol{\sigma}_x^2, \boldsymbol{\mu}_h, \boldsymbol{\sigma}_h^2] \leftarrow \texttt{Linear}(\mathbf{z}_h)$
$\quad$ Get regularization loss $\mathcal{L}_{\text{reg}}$ $\hfill$ (Eq. 5)
**Sample latent code:**
$\quad \mathbf{z} \leftarrow [\boldsymbol{\mu}_x, \boldsymbol{\mu}_h] + [\boldsymbol{\sigma}_x, \boldsymbol{\sigma}_h] \odot \boldsymbol{\epsilon}, \quad \boldsymbol{\epsilon} \sim \mathcal{N}(\mathbf{0}, \mathbf{I})$
**BFN decoding (Alg. 3):**
$\quad \hat{\mathbf{m}} \leftarrow \text{BFNDecode}(\mathbf{z})$
$\quad$ Get reconstruction loss $\mathcal{L}_{\text{recon}}$ $\hfill$ (Eq. 6)
**return** $\mathcal{L}_{recon}, \mathcal{L}_{reg}$

---

### C.3 Bayes Updating Function for Molecular Data

**Continuous coordinates** The receiver observes the noisy input $\mathbf{y}^x$ and the corresponding noise level $\alpha$. Starting from a prior parameterized by $\boldsymbol{\theta}_{i-1}^x = \{\boldsymbol{\mu}_{i-1}, \rho_{i-1}\}$. According to Bayes' rule, it updates its belief by the bayes updating function $h(\boldsymbol{\theta}_{i-1}^x, \mathbf{y}^x, \alpha_i) = (\boldsymbol{\mu}_i, \rho_i)$ :

$$\rho_i = \rho_{i-1} + \alpha_i \tag{24}$$

$$\boldsymbol{\mu}_i = \frac{\rho_{i-1}\boldsymbol{\mu}_{i-1} + \alpha_i \mathbf{y}^x}{\rho_i} \tag{25}$$

**Discrete atom types** Upon receiving the noisy signal $\mathbf{y}^v$ and the noise factor $\alpha_i'$, the receiver updates its belief by applying the Bayes formula with the previous parameters $\boldsymbol{\theta}_{i-1}^v$. The Bayes updating function is:

$$h(\boldsymbol{\theta}_{i-1}^v, \mathbf{y}^v, \alpha_i') := \frac{e^{\mathbf{y}^v} \odot \boldsymbol{\theta}_{i-1}^v}{\sum_{k=1}^K e^{\mathbf{y}_k^v}(\boldsymbol{\theta}_{i-1}^v)_k} \tag{26}$$

where $\odot$ denotes element-wise multiplication.

### C.4 MolFLAE Decoding Process

In the inference phase of BFN, it is in principle possible to draw samples from the receiver distribution to perform Bayesian updates. However, explicitly sampling such intermediate variables—particularly for discrete data—can introduce unnecessary stochasticity and impair the stability of the generation process. Instead, following [26, 49], we directly operate in the parameter space, avoiding noise injection from sampling and enabling a more deterministic and efficient inference.

To implement this approach, we define $\gamma(t) := \frac{\beta(t)}{1-\beta(t)}$. Let $\hat{\mathbf{m}} = [\hat{\mathbf{x}}, \hat{\mathbf{v}}]$ denote the neural network's output at a given step, where $\hat{\mathbf{v}}$ represents the continuous (pre-softmax) logits for atom types. Instead of sampling noisy observations explicitly, we directly use $\hat{\mathbf{m}}$ to update the parameters for the next iteration, thereby bypassing the stochastic sampling step in the standard Bayesian update $\boldsymbol{\theta}_i = h(\boldsymbol{\theta}_{i-1}, \mathbf{y}, \alpha)$.

Under this formulation, the Bayes Flow parameter updates simplify to:

$$p_F\left(\boldsymbol{\mu} \mid \hat{\mathbf{x}}, \mathbf{z}; t\right) = \mathcal{N}\left(\boldsymbol{\mu} \mid \gamma(t)\,\hat{\mathbf{x}},\, \gamma(t)(1-\gamma(t))\,\boldsymbol{I}\right) \tag{27}$$

$$p_F\left(\boldsymbol{\theta}^v \mid \hat{\mathbf{v}}, \mathbf{z}; t\right) = \mathbb{E}_{\mathcal{N}(\mathbf{y}^v \mid \beta(t)(K\mathbf{e}_{\hat{\mathbf{v}}}-\mathbf{1}),\, \beta(t)K\,\boldsymbol{I})}\left[\delta\left(\boldsymbol{\theta}^v - \text{softmax}(\mathbf{y}^v)\right)\right] \tag{28}$$

Here, the continuous coordinates are updated using a closed-form Gaussian posterior, while the discrete atom types are updated via an expected categorical distribution induced by a softmax over noisy logits. In practice, this expectation is approximated with a single Monte Carlo sample.

Throughout the generation process, updates are performed entirely in the parameter space, avoiding noisy sampling steps—except for the final decoding stage, where an actual molecular structure is drawn from the output distribution.

---

**Algorithm 3:** Decoder: sampling Molecules conditioned on latent code

---

**Input:** Network $\mathbf{\Phi}$, latent code $\mathbf{z} \in \mathbb{R}^{N_Z(3+D_Z)}$, total steps $N$, number of atoms $N_M$, number of types $D_M$, noise levels $\sigma_1$, $\beta_1$

**Output:** Sampled molecule $[\hat{\mathbf{x}}, \hat{\mathbf{v}}]$

/* Define update function                                                      */

**Function** update($\hat{\mathbf{x}} \in \mathbb{R}^{N_M \times 3}$, $\hat{\mathbf{v}} \in \mathbb{R}^{N_M \times D_M}$, $\beta(t)$, $\beta'(t)$, $t \in \mathbb{R}^+$)**:**

> $\gamma \leftarrow \frac{\beta(t)}{1-\beta(t)}$
> $\boldsymbol{\mu} \sim \mathcal{N}(\gamma\hat{\mathbf{x}}, \gamma(1-\gamma)\boldsymbol{I})$
> $\mathbf{y}^v \sim \mathcal{N}(\mathbf{y}^v \mid \beta'(t)(D_M \boldsymbol{e}_{\hat{\mathbf{v}}} - 1), \beta'(t)D_M \boldsymbol{I})$
> $\boldsymbol{\theta}^v \leftarrow [\text{softmax}((\mathbf{y}^v)^{(d)})]_{d=1}^{N_M}$
> **return** $\boldsymbol{\mu}, \boldsymbol{\theta}^v$

/* Initialize parameters                                                        */

$\boldsymbol{\mu} \leftarrow 0$, $\boldsymbol{\rho} \leftarrow 1$, $\boldsymbol{\theta}^v \leftarrow \left[\frac{1}{D_M}\right]_{N_M \times D_M}$

**for** $i = 1$ **to** $N$ **do**

> $t \leftarrow \frac{i-1}{n}$
> Sample $\hat{\mathbf{x}}, \hat{\mathbf{v}} \sim p_O(\boldsymbol{\mu}, \boldsymbol{\theta}^v, \mathbf{z}, t)$
> Update latent: $\boldsymbol{\mu}, \boldsymbol{\theta}^v \leftarrow$ update($\hat{\mathbf{x}}, \hat{\mathbf{v}}, \sigma_1, \beta_1, t$)

/* Final sampling                                                           */

Sample $\hat{\mathbf{x}}, \hat{\mathbf{v}} \sim p_O(\boldsymbol{\mu}, \boldsymbol{\theta}^v, \mathbf{z}, 1)$

**return** $[\hat{\mathbf{x}}, \hat{\mathbf{v}}]$

---

# D   Molecules Property Metrics

Following the setup of EDM[15], in our unconditional generation experiments, we employed the following metrics to evaluate the quality of generated molecules:

**Atom Stability**: Proportion of atoms with valid bond counts.

**Molecular Stability**: Proportion of molecules where all atoms are stable.

**Validity**: Proportion of molecules with RDKit-parsable SMILES.

**Novelty**: Proportion of molecules whose SMILES are not in the training set.

**Uniqueness**: Proportion of unique molecules in the generated set.

In our interpolation experiments, to fully assess the chemical and physical properties of intermediary molecules generated during interpolation, we employed the following metrics:

**Similarity Preference**: Defined as follows:

$$\text{Similarity Preference} = \frac{S_t - S_s}{S_t + S_s}$$

where $S_t$ and $S_s$ denote the Tanimoto similarity (calculated using MACCS fingerprints [45]) to the target and source molecules.

**sp3frac**: Represents the proportion of sp3-hybridized carbon atoms in a molecule relative to the total number of carbon atoms.

**BertzCT** [50]: A topological index based on a molecule's structure, considering factors like atomic connectivity, ring size, and number.

**QED** [51]: Evaluates the similarity of a molecule to known drug molecules by considering multiple physicochemical properties.

**Labute ASA** [52]: Measures the surface area of each atom in a molecule in contact with the solvent, reflecting the molecule's solvent interaction ability.

**TPSA** [53]: A value calculated based on a molecule's topological structure and atomic polarity, used to predict solubility and biological membrane penetration.

**logP** [54]: The logarithm of the partition coefficient of an organic compound between octanol and water, indicating the hydrophobicity of a molecule.

**MR** [54]: Measures a molecule's ability to refract light, related to factors like polarizability, molecular weight, and density.

# E  Results on drug-likeness metrics on GEOM-Drugs

To provide a more comprehensive evaluation, we conduct experiments on the GEOM-Drugs dataset with additional drug likeness metrics, where MolFLAE showed significant improvements over existing baselines. The results are summarized in Table 7. It should be noted that we were unable to locate open resources for GeoBFN's checkpoints on GEOM-Drugs, so experiments on GeoBFN were not conducted.

In details, we sample 10,000 molecules from each model with their default settings, obtaining atom positions and types, and then inferred bond types using OpenBabel. We then fix the bond order using Schordinger due to some bugs in OpenBabel. The final molecules are then be evaluated using RDKit for the following metrics: QED, SA, Lipinski, and Strain Energy.

Table 7: Comparison of models on QED, SA, Lipinski and Strain Energy on GEOM-Drugs dataset.

| # Metrics | QED ($\uparrow$) | SA ($\uparrow$) | Lipinski ($\uparrow$) | Strain Energy ($\downarrow$) |
|---|---|---|---|---|
| Data | 0.64 | 0.84 | 4.80 | 80.19 |
| EDM | 0.36 | 0.59 | 4.26 | 705.2 |
| GeoLDM | 0.40 | 0.63 | 4.31 | 446.1 |
| UniGEM | 0.36 | 0.63 | 4.24 | 490.4 |
| MolFLAE | **0.60** | **0.75** | **4.75** | **84.66** |

# F  Ablation Study

We conduct ablation experiments on three key design choices of MolFLAE: the presence of the regularization loss, the feature embedding dimension $D_Z$, and the number of virtual nodes $N_Z$. All models are evaluated using 100-step decoding. Unless otherwise noted, all hyperparameters follow the configuration in Table 10.

Table 8 summarizes the results on QM9. Each row varies only one component while keeping all other settings fixed.

We do ablation on Regularization loss, embedding dimension $D_Z$ and length of latent codes $N_Z$. Our decoder uses 100 steps for sampling. And the MolFLAE means the same hyperparameters as in Table 10. The following three experiments keeps the same settings except for the marked one. Removing the regularization loss leads to a drop in molecular stability and novelty, suggesting that

Table 8: Ablation study, performance comparison of different model config on QM9.

| Model Config (steps=100) | Atom Sta (%) | Mol Sta (%) | Valid (%) | V×U (%) | Novelty (%) |
|---|---|---|---|---|---|
| MolFLAE | **99.39** | **92.01** | 96.81 | 88.94 | **74.49** |
| w/o Regularization Loss | 98.73 | 85.01 | **97.82** | 86.43 | 66.04 |
| $D_Z = 16$ | 98.90 | 87.05 | 93.09 | **94.19** | 70.32 |
| $N_Z = 5$ | 99.21 | 89.24 | 96.60 | 73.21 | 61.54 |

latent smoothness is crucial for robust generation. Reducing the latent dimensionality ($D_Z = 16$)

or the number of latent nodes ($N_Z = 5$) also impacts overall performance, particularly in terms of novelty and reconstruction fidelity.

Moreover, as our iterative decoding process is a little complex, we tried to simplify it to a one-step decoding variant. However, this approach failed to generate any valid molecules.

## G Hyper-parameter Settings

Hyperparameters for training on QM9, GEOM-DRUG and ZINC-9M are listed in Table 10. We followed prior works in the choice of network structure, and carried out ablation study to determine the number of latent codes introduced.

Table 9: Training costs.

| Dataset | GPUs | Time | Max Epoch |
|---------|------|------|-----------|
| GEOM-DURG | 4 Nvidia A100s(80G) | 6 days | 15 |
| QM9 | 4 Nvidia A100s(80G) | 16h | 250 |
| ZINC-9M | 8 Nvidia A800s(80G) | 3 days | 25 |

Table 10: Hyperparameters for training.

| Parameter | Value or description |
|-----------|----------------------|
| Train/Val/Test Splitting | 6921421/996/remaining data for GEOM-DRUG
9322660/932/remaining data for ZINC-9M
100000/17748/remaining data for QM9 |
| Batch size | 100 for GEOM-DRUG,200 for ZINC-9M,400 for QM9 |
| Optimizer | Adam |
| $\beta_1$ | 0.95 |
| $\beta_2$ | 0.99 |
| Lr | 0.005 |
| Weight decay | 0 |
| Learning rate decay policy | ReduceLROnPlateau |
| Learning rate factor | 0.4 for GEOM-DRUG, 0.6 for QM9 and ZINC-9M |
| Patience | 3 for GEOM-DRUG, 10 for QM9 and ZINC-9M |
| Min learning rate | 1.00E-06 |
| Embedding dimension $D_f$ | 128 |
| Head number | 16 |
| Layer number | 9 |
| $k$ (knn) | 32 |
| Activation function | ReLU |
| $N_Z$ | 10 |
| $D_Z$ | 32 |
| $\text{var}_x$ | 100 |
| $\text{var}_h$ | 1 |
| Reconstruction loss weight | 1 |
| Regularization loss weight | 0.1 |

## H Broader Impacts and Safety Discussion

Our work develops an autoencoder model for molecular design, which has potential positive societal impact in areas such as drug discovery, materials science, and green chemistry by enabling the

efficient generation of candidate molecules with desired properties. However, we acknowledge that it may also be misused, for instance to generate harmful or toxic compounds. While our model is trained and evaluated on general-purpose datasets without any bias toward hazardous compounds, we emphasize that any downstream deployment should include domain-specific safeguards, such as toxicity filters and expert oversight.

