# OpenReview forum: "Manipulating 3D Molecules in a Fixed-Dimensional E(3)-Equivariant Latent Space"
_NeurIPS.cc/2025/Conference — NeurIPS 2025 poster_

### Official Review · Reviewer_fHNV · 2025-07-02

**Clarity:** 4
**Significance:** 4
**Originality:** 3
**Rating:** 5
**Confidence:** 4

**Summary:**

This paper presents MolFLAE, an SE(3)-equivariant variational autoencoder (VAE) designed for 3D molecular structures. To ensure a fixed-dimensional latent representation, it introduces a fixed number of virtual nodes. A Bayesian Flow Network is then used to reconstruct molecules from these latent codes.

MolFLAE is evaluated across a variety of tasks, demonstrating its versatility in applications such as:
- Unconditional molecular generation
- Analog generation with varying atom counts
- Mixing invariant and equivariant codes of distinct molecules
- Interpolating between source and target molecules
- Optimizing drug candidates for hGR
The model also shows promise for broader molecular applications, including molecule inpainting and structural alignment.

**Questions:**

1. Have the authors considered extending the model with a diffusion process in the latent space (i.e., using a latent diffusion model)? I’d be interested to hear the authors’ thoughts on how this might improve performance.
2. It seems that molecular validity is reported only for the unconditional generation task. I may have missed it, but were validity (and/or stability) checks also performed for the downstream applications?
3. Beyond the noted issues with disentanglement and interpretability, are there other limitations the authors would like to highlight?

**Ethical Concerns:**

["NO or VERY MINOR ethics concerns only"]

**Final Justification:**

The authors have thoroughly addressed my concerns, so I am keeping my acceptance score. However, I did not raise it to 6 due to current limitations, such as the need for atom count during decoding and limited scalability to larger molecules.

**Limitations:**

Yes

**Quality:**

3

**Strengths And Weaknesses:**

# Strengths
- The paper makes a good technical contribution by introducing a more effective method to capture rich molecular features while maintaining a fixed-dimensional latent representation. Instead of relying on simple pooling, it introduces multiple virtual nodes to achieve this.
- It demonstrates the versatility of MolFLAE through a wide range of molecular tasks, addressing a key limitation of prior models that were typically designed for narrow or specific subtasks.

# Weaknesses
- According to Appendix A.1, the authors appear to use E(3)-equivariance rather than SE(3)-equivariance, which may limit the model's ability to capture chirality. Please correct me if I’ve misunderstood this point.
- While the paper offers novel contributions, certain components are similar to prior work, such as UniMoMo [1] and PepGLAD [2], which also model the molecules with E(3)-equivariant VAE. Although MolFLAE may be the first to apply these ideas to 3D small molecules, acknowledging related efforts in other molecular domains would help better contextualize the work and strengthen its positioning.
- (Minor) While I'm familiar with the molecular domain and the techniques used, the evaluation metrics for 3D small molecules are less familiar. A brief explanation -- e.g., in Appendix D -- of how metrics like atom stability, molecular stability, validity, and novelty are defined would be helpful.

## Typos
- Line 107: inculding -> including
- Line 191: Alogorithm -> Algorithm
- Line 192: modelities -> modalities
- Line 227-228: Shouldn’t $z_h^1$ correspond to the substructure information?
- Line 574: Incomplete sentence: “And pass the Neural Network”
- Equation (12) in Appendix B: The equation $[\sigma_x^2, \mu_h, \sigma_h^2] = \text{Linear}([z_x, z_h])$ appears misleading, as the actual implementation is $[\sigma_x^2, \mu_h, \sigma_h^2] = \text{Linear}(z_h)$, without dependence on $z_x$, in order to preserve SE(3)-equivariance.

## References
[1] Kong, Xiangzhe, et al. "UniMoMo: Unified Generative Modeling of 3D Molecules for De Novo Binder Design." arXiv preprint arXiv:2503.19300 (2025).
[2] Kong, Xiangzhe, et al. "Full-atom peptide design with geometric latent diffusion." arXiv preprint arXiv:2402.13555 (2024).

---

> ### Author Rebuttal · Authors · 2025-07-30
>
> Dear reviewer fHNV,
>
> Thank you for pointing out the typos to help us improve the quality of our work. We sincerely apologize for these oversights. We will fix them in the camera-ready version. We respond to your concerns one by one.
> >W1:  According to Appendix A.1, the authors appear to use E(3)-equivariance rather than SE(3)-equivariance, which may limit the model's ability to capture chirality. Please correct me if I’ve misunderstood this point.
>
> Thank you for pointing out this important detail. Our model is E(3)-equivariant, and a careful examination of the proof confirms that it does not require the orthogonal transformation to have determinant 1. We acknowledge that referring to it as SE(3)-equivariant throughout the paper may be misleading. We will revise this terminology in the camera-ready version.
>
> From a mathematical perspective, because SE(3) is a subgroup of E(3), any E(3)-equivariant model is also SE(3)-equivariant. We followed prior works like MolCRAFT [1] and TargetDiff [2] in using SE(3), since we adopt the same network architectures.
>
> Importantly, this distinction does not affect the correctness of our results. E(3)-equivariance is more suitable for generative tasks like molecule editing, as it preserves the chirality of latent codes and molecules. SE(3)-only equivariance (i.e., excluding reflections) would be more appropriate for chirality-sensitive property prediction, which is beyond our current scope.
>
> Moreover, our main baselines including GeoBFN[3], EDM[4], and UniGEM[5] are also based on E(3)-equivariant networks. Thus, our comparisons are on fair ground.
>
> [1] Yanru Qu et al, MolCRAFT: Structure-based drug design in continuous parameter space. ICML, 2024.
>
> [2] Jiaqi Guan et al, 3D Equivariant Diffusion for Target-Aware Molecule Generation and Affinity Prediction. ICLR, 2023.
>
> [3] Yuxuan Song et al, Unified generative modeling of 3d molecules with bayesian flow networks. ICLR, 2024.
>
> [4] Emiel Hoogeboom et al,  Equivariant diffusion for molecule generation in 3d. ICML, 2022.
>
> [5] Shikun Feng et al,  UniGEM: A unified approach to generation and property prediction for molecules. ICLR, 2025.
>
>
>
> ---
> >W2: While the paper offers novel contributions, certain components are similar to prior work, such as UniMoMo [1] and PepGLAD [2], which also model the molecules with E(3)-equivariant VAE. Although MolFLAE may be the first to apply these ideas to 3D small molecules, acknowledging related efforts in other molecular domains would help better contextualize the work and strengthen its positioning.
>
> We thank the reviewer for raising this important point. While MolFLAE shares high-level similarities with UniMoMo [1] and PepGLAD [2] in leveraging E(3)-equivariant VAEs, our approach differs significantly in motivation, architecture, and application scope.
>
> We have already discussed PepGLAD in the related work section under Unconditional Generation for 3D Molecules, and we will include UniMoMo in the camera-ready version as they are quite similar. Both UniMoMo and PepGLAD compress each molecular block (e.g., an amino acid residue) into a single latent node. This design has two key limitations. First, the number of latent nodes depends on the molecular composition, making cross-sample comparisons and operations like interpolation difficult. Second, the spatial relationships between molecular blocks are tightly preserved, restricting the flexibility of the latent space for generative modeling, e.g. when latent codes are used for autoregressive modeling by finetuning LLMs.
>
> In contrast, MolFLAE encodes entire molecules into a fixed-length 1D latent sequence, where virtual nodes are associated with a learnable embedding to assign an intrinsic order. This greatly facilitates applications such as molecular interpolation and molecule-to-molecule translation (optimization).
>
> In comparison with current vision-language models, MolFLAE adopts a CLIP-style encoder with a diffusion-like decoder, akin to recent advances in vision-language models (e.g., Emu2 [3] and BLIP3-o[4]), while UniMoMo or PepGLAD use pixel-level reconstructions. Our latent codes are not atom-aligned but rather semantically rich and highly compressed, similar to how CLIP-style models encode images. This makes our approach more suitable for downstream semantic tasks and aligns with the emerging paradigm in multimodal generation. This paradigm has been proven to be powerful in transforming non-1D modalities like images into 1D tokens.
>
> We will ensure these distinctions are more clearly highlighted in the related work part of the camera-ready version.
>
> [1] Kong, Xiangzhe, et al. "UniMoMo: Unified Generative Modeling of 3D Molecules for De Novo Binder Design."  arXiv:2503.19300 (2025).
>
> [2] Kong, Xiangzhe, et al. "Full-atom peptide design with geometric latent diffusion." arXiv:2402.13555 (2024).
>
> [3] Sun, Quan, et al. "Generative multimodal models are in-context learners." CVPR, 2024.
>
> [4] Chen, Jiuhai, et al. "Blip3-o: A family of fully open unified multimodal models-architecture, training and dataset." arXiv:2505.09568 (2025).
>
>
> ---
> >W3: While I'm familiar with the molecular domain and the techniques used, the evaluation metrics for 3D small molecules are less familiar. A brief explanation -- e.g., in Appendix D -- of how metrics like atom stability, molecular stability, validity, and novelty are defined would be helpful.
>
> Thank you very much for your insightful comments. To be self-contained, we will extend Appendix D in the camera-ready version.
> Here is a precise metric definition (all in [0,1]):
> - Atom Stability: Proportion of atoms with valid bond counts.
> - Molecular Stability: Proportion of molecules where all atoms are stable.
> - Validity: Proportion of molecules with RDKit-parsable SMILES.
> - Novelty: Proportion of molecules whose SMILES are not in the training set.
>
> ---
> >Q1:  Have the authors considered extending the model with a diffusion process in the latent space (i.e., using a latent diffusion model)? I’d be interested to hear the authors’ thoughts on how this might improve performance.
>
> Thank you for the insightful question. We agree that extending MolFLAE with a latent diffusion model is a promising direction. Since our model provides a high-quality molecule decoder, the generation task can be naturally framed as generating latent codes. This offers an opportunity for controllable molecule design with improved validity and structural fidelity, especially because the decoder has been trained on more diversified molecules than typical conditional molecular generation datasets.
>
> Furthermore, your suggestion opens up exciting avenues for future work. For example, integrating latent diffusion or latent flow matching could enable more precise and diverse molecular edits. These techniques may also facilitate conditional generation with strong generalization.
>
> We appreciate your valuable suggestion and will actively consider this line of research in future iterations of our work.
>
> ---
> >Q2: It seems that molecular validity is reported only for the unconditional generation task. I may have missed it, but were validity (and/or stability) checks also performed for the downstream applications?
>
> Thank you very much for your careful observation regarding the validity checks. We sincerely apologize for not including them in our initial submission. To address this, we have conducted additional experiments to evaluate the validity and stability of all molecules involved in the three downstream tasks. Here are the detailed results:
> ### Generating Analogs with Different Atom Numbers (Figure 2)
> |Atom Number|-2|-1|0|1|2|
> |-|-|-|-|-|-|
> |Valid(%)|100.0|99.89|99.76|99.89|99.68|
> |Atom Sta(%)|84.58|83.28|82.48|82.38|82.53|
>
> ### Exploring the disentanglement of the latent space via molecule reconstruction (Figure 3)
> ||$(z_{h_{0}},z_{x_{0}})$|$(z_{h_{1}},z_{x_{1}})$|$Preserving\ z_{x}$|$Preserving\ z_{h}$|
> |-|-|-|-|-|
> |Valid(%)|100.0|100.0|100.0|100.0|
> |Atom Sta(%)|80.90|80.81|85.20|84.62|
>
> ### Latent Interpolation (Table 4 & 5)
> ||Start|2|3|4|5|6|7|End|
> |-|-|-|-|-|-|-|-|-|
> |Valid(%)|99.90|99.90|99.90|100.0|99.90|100.0|100.0|100.0|
> |Atom Sta(%)|82.23|82.95|84.17|84.57|84.21|83.74|83.10|82.63|
>
> ||Start|2|3|4|5|6|7|8|9|End|
> |-|-|-|-|-|-|-|-|-|-|-|
> |Valid(%)|100.0|99.90|99.90|99.80|99.80|100.0|100.0|100.0|99.80|99.90|
> |Atom Sta(%)|82.23|82.75|83.53|84.07|84.36|84.35|84.16|84.17|82.78|82.67|
>
> ||Start|2|3|4|5|6|7|8|9|10|11|End|
> |-|-|-|-|-|-|-|-|-|-|-|-|-|
> |Valid(%)|99.90|100.0|99.90|100.0|100.0|100.0|100.0|100.0|99.90|99.90|100.0|99.80|
> |Atom Sta(%)|82.39|82.50|83.76|83.67|83.92|84.65|84.20|84.50|84.27|83.00|82.84|82.57|
>
> The validity is high among all experiments. Thank you once again for your valuable feedback and for bringing this to our attention.
>
> ---
> >Q3: Beyond the noted issues with disentanglement and interpretability, are there other limitations the authors would like to highlight?
>
> We appreciate the reviewer’s suggestion to reflect on further limitations. We highlight the following points:
> 1. Atom Count Specification During Decoding: Our current decoder requires specifying the number of atoms. Although this is common in diffusion-based molecule generation, it could be eliminated in our setting by a small atom count predictor conditioned on the latent code.
> 2. Broader Validation of Editing Capabilities Needed: Figure 6 shows preliminary editing results, but we have not yet systematically evaluated MolFLAE across diverse editing tasks. We plan to explore this in future work.
> 3. Scalability to Large Molecules: Our current experiments focus on small to medium-sized molecules. Extending the framework to larger structures such as proteins is a promising but non-trivial extension.
>
>
> ---

---

> ### Comment · Reviewer_fHNV · 2025-08-05
>
> Thank you for your detailed response to my questions and comments. My concerns have been addressed. I am pleased to maintain my score for acceptance.

---

> ### Author Response · Authors · 2025-08-05
>
> Thanks again for your insightful discussion with us. We're glad to hear you're satisfied with our response.

---

### Official Review · Reviewer_EbXx · 2025-07-02

**Clarity:** 3
**Significance:** 2
**Originality:** 3
**Rating:** 4
**Confidence:** 2

**Summary:**

This paper introduces a novel VAE architecture for learning 3D molecules. It uses learnable virtual nodes to achieve a fixed number of latent nodes. Thus, MolFLAE has an SE(3)-equivariant latent space with fixed dimension independent from the number of atoms, allowing for easier interpolation or extrapolation. The decoder of architecture employs a Bayesian Flow Network. The author evaluated MolFLAE on various downstream tasks, demonstrating model efficacy. The MolFLAE also exhibits the ability to learn disentangled spatial and semantic latent components.

**Questions:**

1. In formula (3), the model discards the embedding of molecule M and only retains the embedding of virtual nodes Z. Since the number of virtual nodes is fixed-length, will the performance of the model vary for molecules with different numbers of atoms, particularly for very large molecules?
2. In the GEOBFN paper, the best performance was achieved when sampling the molecules with 2000 steps. Did the author try to compare the performance of MolFLAE when sampling with 2000 steps?
3. In drug discovery, it is crucial to generate drug-like molecules. Why are metrics like drug-likeness (QED) not used to evaluate the model with the GEOM-drug dataset?

**Ethical Concerns:**

["NO or VERY MINOR ethics concerns only"]

**Final Justification:**

The author has addressed all my questions. The quality of the work meets the standard for a NeurIPS acceptance. However, I still believe the performance boost is a little bit marginal, and I am not completely satisfied with the significance of the work. Thus, I will retain my score as it is.

**Limitations:**

Yes.

**Quality:**

3

**Strengths And Weaknesses:**

**Strengths**

1. It is a common problem in molecular VAE where the number of atoms varies across molecules. The author provides a novel approach by introducing virtual nodes to effectively achieve a fixed-dimensional latent space.
2. The authors demonstrate that the MolFLAE can generate similar molecules with the same latent code, despite varying numbers of atoms, thereby illustrating the smoothness of model generation. The disentanglement of spatial and semantic features also shows the possibility of generating molecules with desired properties while maintaining certain constraints.
3. The paper demonstrated the model's performance in optimizing molecules targeting the human glucocorticoid receptor (hGR). This real-world task shows the effectiveness of the model in drug design applications.

**Weaknesses**
1. Table 1 shows that MolFLAE outperforms baselines across different metrics. However, it seems the improvement is a bit marginal. The author should have more comparisons and discussions between MolFLAE and baselines other than unconditional molecule generation.

---

> ### Author Rebuttal · Authors · 2025-07-30
>
> Many thanks to your valuable comments and questions, which help us a lot to improve our work. We address your questions as follows.
> >W1.1:  Table 1 shows that MolFLAE outperforms baselines across different metrics. However, it seems the improvement is a bit marginal.
>
> Thank you very much for your insightful comments. We truly appreciate your thoughtful feedback and suggestions.
>
> We acknowledge that while MolFLAE demonstrates improvements over baseline models in Table 1, these enhancements may appear relatively modest. This is likely because the QM9 tasks are relatively simple. To provide a more comprehensive evaluation, we conducted experiments on the GEOM-Drugs dataset with additional drug likeness metrics, where MolFLAE showed significant improvements over existing baselines. We sincerely appreciate your Question 3 about drug likeness, which inspires us to do these experiments.
> |  Model & sampling steps | QED(↑)   | SA(↑)    | Linpinski(↑) | Median Strain Energy(↓) |
> | ------------- | ------ | ------ | ---------- | --------------------- |
> | data| 0.64 | 0.84 | 4.80     | 80.19|
> | EDM 1000| 0.36 | 0.59 | 4.26     | 705.2|
> | GeoLDM 1000| 0.40 | 0.63 | 4.31     | 446.1|
> | UniGEM 1000 | 0.36 | 0.63 | 4.24     | 490.4|
> | MolFLAE 100 | **0.60** | **0.75** | **4.75**     | **84.66**|
>
> It should be noted that we were unable to locate open resources for GeoBFN's checkpoints on GEOM-Drugs, so experiments on GeoBFN were not conducted. We will provide the detailed results of these experiments in the camera-ready version.
>
> Also, it is important to clarify that our primary objective is not to build a state-of-the-art unconditional generative model. Instead, we aim to demonstrate that a fixed-dimensional, equivariant latent space can effectively capture molecular semantics. This capability is crucial for a wide range of downstream tasks, such as molecular modification and optimization.
>
> >W1.2 The author should have more comparisons and discussions between MolFLAE and baselines other than unconditional molecule generation.
>
> To further contextualize our results, we provide an additional comparison against several recent structure-based drug design (SBDD) models on standard drug-like property metrics including QED, SA and Lipinski score, as shown below. These results indicate that our model could serve as a strong decoder for latent diffusion models in the field of SBDD, providing better drug likeness and synthesizability.
> | Model                        | QED↑     | SA↑      | Linpinski↑|
> |-----------------------------|----------|----------|------------|
> | TargetDiff（from UniMoMo ） | 0.49     | 0.60     | 4.57       |
> | MolCRAFT（from UniMoMo ）   | 0.48     | 0.66     | 4.39       |
> | UniMoMo                     | 0.55     | 0.70     | 4.68      |
> | MolFLAE（100 steps）        | **0.60** | **0.75** | **4.75** |
>
> However, we respectfully note that direct comparisons to conditional generation models may not be entirely appropriate, as they often operate on different datasets.
>
> Thank you again for your valuable feedback. We are committed to improving this work and greatly appreciate your guidance.
>
>
> ---
> >Q1:  In formula (3), the model discards the embedding of molecule M and only retains the embedding of virtual nodes Z. Since the number of virtual nodes is fixed-length, will the performance of the model vary for molecules with different numbers of atoms, particularly for very large molecules?
>
> Thank you for your question regarding the potential performance variation of our model for molecules with different numbers of atoms, especially large molecules. We conduct additional experiments to evaluate reconstruction similarity and validity across different heavy atom counts. The results are summarized in the table below:
>
> | Heavy Atom Count         | [0, 20) | [20, 30) | [30, 40) | [40, 50) | [50, +∞) |
> |--------------------------|---------|----------|----------|----------|----------|
> | Averaged Similarity(%)   | 94.16  | 88.62   | 73.43   | 65.99   | 49.30   |
> | Valid(%)                  | 99.19  | 99.80   | 100.0   | 100.0   | 100.0   |
> | Proportion in Train Set(%)| 12.17  | 58.43   | 26.39   | 2.89   | 0.12   |
>
> We observe that larger molecules tend to be reconstructed with lower similarity. This is expected for two reasons:
> 1. The training set contains very few large molecules, making it difficult for the model to learn their distribution.
> 2. As shown in our ablation study (Table 6 in Appendix E), the number of virtual nodes would slightly affect modeling performance. Larger molecules require more virtual nodes for accurate reconstruction.
>
> In Computer Vision (CV), extensive compression is often feasible [1,2,3], but it typically relies on a strong decoder. In contrast, our decoder is relatively lightweight, a 9-layer Transformer with only 2.83M parameters. Our model is a prototypical design rather than a fully optimized solution. While modeling large molecules remains a critical and challenging task, we believe our framework provides a promising foundation, and we are keen to explore its extension in this direction in future work.
>
> [1] Qihang Yu et al, An image is worth 32 tokens for reconstruction and generation. NeurIPS, 2024.
>
> [2] Chen, Jiuhai, et al. "Blip3-o: A family of fully open unified multimodal models-architecture, training and dataset." arXiv:2505.09568 (2025).
>
> [3] Sun, Quan, et al. Emu: Generative pretraining in multimodality. ICLR, 2024.
>
> ---
> >Q2: In the GEOBFN paper, the best performance was achieved when sampling the molecules with 2000 steps. Did the author try to compare the performance of MolFLAE when sampling with 2000 steps?
>
> Thank you very much for your question regarding the sampling steps in our work and the comparison with GEOBFN.
>
> We have evaluated the performance of MolFLAE across various sampling steps, including 500, 1000, and 2000 steps, and compared it with GEOBFN’s performance at 2000 steps.
>
> In our experiments, we found that when MolFLAE was sampled at just 100 steps, its performance was already comparable to that of GEOBFN at 2000 steps and significantly outperformed GEOBFN's results at 100 steps. However, while increasing the sampling steps did lead to improvements in Atom Stability, Mol Stability, and Validity, these improvements were rather marginal, and we noticed a decline in Uniqueness and Novelty.
> | Model & sampling steps | Atom Sta(%)      | Mol Sta(%)| Valid(%)| U×V(%)| Novelty(%)   |
> |------------------------|------------------|----------------|----------------|----------------|-----------|
> | Geobfn 100             | 98.64       | 87.21    | 93.03    | 91.53    | 70.3      |
> | Geobfn 2000            | 99.31            | 93.32          | 96.88          | **92.41**      | 65.3      |
> | MolFLAE 100            | 99.39            | 92.01          | 96.81          | 88.94          | **74.49** |
> | MolFLAE 500            | 99.58            | 94.52          | 98.18          | 80.17          | 68.86     |
> | MolFLAE 1000           | **99.61**        | 94.81          | 98.38          | 75.57          | 71.05     |
> | MolFLAE 2000           | **99.61**        | **94.99**      | **98.49**      | 72.47          | 65.37     |
>
> These observations and the balance between computation efficiency and performance, combined with the fact that our original goal was not to develop a powerful unconditional generation model but rather to create a model capable of flexible molecular modification, led us to present the 100-step results in our paper. The 100-step results effectively demonstrated the model's capabilities without necessitating a large number of sampling steps.
>
>
>
> ---
> >Q3: In drug discovery, it is crucial to generate drug-like molecules. Why are metrics like drug-likeness (QED) not used to evaluate the model with the GEOM-drug dataset?
>
> Thank you for bringing up the important point regarding the evaluation metrics used in our model with the GEOM-Drugs dataset.
>
> In our experiments with the GEOM-Drugs dataset, we initially followed the setup of EDM, GeoLDM and UniGEM, which primarily focused on testing Validity and Atom Stability. In response to your inquiry, we have further evaluated our model as well as these three baselines with additional metrics, including drug-likeness (QED), synthetic accessibility and Lipinski's rule of five, where we significantly outperform baseline methods. Detailed results are in answer to W1.1.
>
> We sincerely hope this information addresses your concerns and provides a clearer picture of our model's performance. Thank you once again for your valuable feedback.
>
> ---

---

### Official Review · Reviewer_dc5U · 2025-07-07

**Clarity:** 3
**Significance:** 4
**Originality:** 4
**Rating:** 4
**Confidence:** 4

**Summary:**

This paper introduces a VAE for 3D molecule generation with a fixed-size equivariant latent space. It allows flexible manipulation of molecules in a zero-shot way. The model shows good results on generation, editing, and drug optimization tasks. The fixed-length latent space enables interpolation and disentanglement.

**Questions:**

-- How sensitive is the method to the number of virtual nodes?
-- Could the learned latent space be used for other molecular tasks? Which ones?
-- Are the interpolation results robust across different dimensions of variation (molecular types, etc)?

**Ethical Concerns:**

["NO or VERY MINOR ethics concerns only"]

**Final Justification:**

I thank the authors for a good response.
The response further supports my evaluation.

**Limitations:**

yes

**Paper Formatting Concerns:**

/

**Quality:**

4

**Strengths And Weaknesses:**

Strengths:
-- Novel latent space design for SE(3)-equivariant molecular representations.
-- Strong experimental results on several datasets.
-- Demonstrates useful molecule editing tasks (e.g., atom count change, interpolation).
-- Includes a real drug design use case with docking evaluation.

Weaknesses
-- Latent disentanglement is only partial and not deeply explored. Doing more here would be very intereting.
-- Method is complex; requires many architectural components. Can it be simplified and better ablated?
-- Limited comparison to recent unified editing models. Would really like to see more here.

---

> ### Author Rebuttal · Authors · 2025-07-30
>
> We thank the reviewer for the thoughtful and encouraging feedback.  Below, we address your concerns and questions.
>
> >W1:  Latent disentanglement is only partial and not deeply explored. Doing more here would be very interesting.
>
> We appreciate the reviewer’s insightful observation regarding the extent of latent disentanglement. Our current model does not incorporate any explicit supervision signal or inductive bias specifically enforcing disentanglement between invariant and equivariant features. While partial disentanglement emerges naturally from our architecture and training objective, we agree that further exploration would be valuable. One promising direction is to employ a contrastive learning objective that encourages consistency of the invariant representation across different conformations of the same molecule. This could explicitly encourage the model to separate invariant and equivariant features more effectively. We thank the reviewer for highlighting this important area and will pursue this line of investigation in future work.
>
>
> ---
> >W2: Method is complex; requires many architectural components. Can it be simplified and better ablated?
>
> We thank the reviewer for pointing out the complexity of the method. Our approach is fundamentally built upon a standard VAE framework, consisting of an encoder, a latent space with regularization, and a decoder. The main architectural difference lies in the decoder, where we introduce an iterative decoding mechanism (Bayesian flow network, BFN) similar to diffusion models. This modification is critical for generating valid molecular structures. We experimented with a simplified, one-step decoding variant (i.e., without iterative decoding), but it failed to produce any valid molecules. Due to the poor quality and instability of this baseline, we excluded it from the main ablation results to maintain focus on more informative comparisons. Nonetheless, we acknowledge that including such a baseline, even as a failure case, could be instructive, and we will consider adding it in the appendix for completeness. In summary, the current architecture reflects the minimal design necessary to achieve effective and valid molecular generation, and we are actively exploring ways to further streamline the components without compromising performance.
>
>
> ---
> >W3: Limited comparison to recent unified editing models. Would really like to see more here.
>
> Thank you for requiring more discussion on other unified editing models. To the best of our knowledge, MolFLAE is the first fixed-length equivariant molecular autoencoder capable of unified latent space manipulation for 3D molecules. This includes zero-shot operations like atom editing, structure inpainting, and property-guided interpolation, all within the same latent space.
>
> A close prior work is the SMILES-based VAE [1], which enables continuous latent space editing but lacks any modeling of 3D structure or equivariant properties. Another recent approach [2] introduces a fixed-length autoencoder but relies on global pooling over atom features, discarding spatial and geometric details critical for reconstruction. UAE-3D [3] has proposed a 3D fixed-length latent space by discarding the inductive bias of geometric equivariance. Furthermore, both [2] and [3] adopt evaluation settings different from ours and do not provide open-source code, making direct comparison difficult.
>
> While most prior editing methods target specific tasks (e.g., linker design [4], scaffold inpainting [5], and Gradient-guided or property-based optimization [6,7]), they are not designed as general-purpose frameworks. Notably, MolFLAE can handle such tasks under a unified interface (e.g., see linker design and superposition in Figure 6), and we will explore broader evaluations in our future works.
>
> In summary, we struggled to find editing models with the same settings as ours for a fair comparison. MolFLAE is a powerful Auto-Encoder, which means a broader application to other editing tasks.
>
> [1]Gómez-Bombarelli et al, “Automatic Chemical Design Using a Data-Driven Continuous Representation of Molecules.” ACS Central Science 4 (2): 268–76, 2018.
>
> [2]Tianxiao Li, Martin Renqiang Min, Hongyu Guo, and Mark Gerstein. 3d autoencoding diffusionmodel for molecule interpolation and manipulation, 2024.
>
> [3]Yanchen Luo, Zhiyuan Liu, Yi Zhao, Sihang Li, Kenji Kawaguchi, Tat-Seng Chua, and XiangWang. Towards unified latent space for 3d molecular latent diffusion modeling, 2025.
>
> [4]Ilia Igashov et al, Equivariant 3d-conditional diffusion models for molecular linker design, 2022.
>
> [5]Shicheng Chen et al, Deep lead optimization enveloped in protein pocket and its application in designing potent and selective ligands targeting LTK protein. Nat. Mac. Intell., 7(3):448–458, 2025.
>
> [6]Keyue Qiu et al, Empower structure-based molecule optimization with gradient guidance, 2025.
>
> [7]Alex Morehead and Jianlin Cheng. Geometry-complete diffusion for 3d molecule generation and optimization. ArXiv, 2023.
>
>
>
> ---
> >Q1:  How sensitive is the method to the number of virtual nodes?
>
> We appreciate the reviewer’s question regarding the sensitivity of our method to the number of virtual nodes. We investigated this by reducing the number of virtual nodes ($N_Z$) from 10 to 5 on the QM9 dataset. As shown in Table 6 (Appendix E) in our paper, this setting yields a molecular stability of 89.24% and validity of 96.60%. While slightly lower than the performance with 10 nodes, these results are still strong and indicate that the method is relatively robust to moderate changes in the number of virtual nodes. Lower numbers of virtual nodes correspond to a smaller latent space capacity, which may restrict the diversity of generated molecules. As a result, the model tends to produce outputs more similar to the training data, potentially leading to lower novelty scores. We show Table 6 here for convenience:
> | Model Config (steps=100)|Atom Sta (%)|Mol Sta (%)|Valid (%)|V×U (%)|Novelty (%)|
> |-|-|-|-|-|-|
> |**MolFLAE**|**99.39**|**92.01**|96.81|88.94|**74.49**|
> |w/o Regularization Loss| 98.73| 85.01| **97.82** | 86.43| 66.04|
> |$D_Z=16$|98.90|87.05|93.09|**94.19**|70.32|
> |$N_Z=5$|99.21|89.24|96.60|73.21|61.54|
>
> We acknowledge that more virtual nodes may be required for modeling larger and more complex molecular structures, such as peptides or proteins, due to the inherent compression capacity of limited virtual nodes. However, recent work in image generation, such as TiTok [1], demonstrates that even high-resolution images can be compressed into extremely compact latent representations (e.g., 32 tokens) while preserving generation quality. These results suggest promising scalability potential for our approach as well, motivating future extensions toward more efficient tokenization strategies for large biomolecules.
>
> [1] Qihang Yu et al, An image is worth 32 tokens for reconstruction and generation. NeurIPS, 2024.
>
> ---
> >Q2: Could the learned latent space be used for other molecular tasks? Which ones?
>
> Thank you for this thoughtful question. Yes, the learned latent space in our model offers promising utility beyond unconditional generation and can potentially support a range of downstream molecular tasks:
> 1. Conditional Generation: By applying diffusion or flow-matching techniques in the latent space, we can perform conditional latent code generation with given pocket structures. This could enable controllable design with high validity and structural fidelity, because our decoder is trained on much more diversified molecules than the current conditional molecular generation datasets.
> 2. Property Prediction: The compact invariant latent representations can be used as input features for supervised property prediction tasks, acting as learned molecular embeddings that capture both global and local structural information.
> 3. Language-Guided Molecular Generation: Perhaps the most exciting direction is leveraging the 1D latent codes as a “molecular language” compatible with large language models (LLMs). This opens the door to natural language-driven molecular generation, akin to the recent BAGEL model [1] or BLIP3-o [2], which integrates diverse visual modalities into pretrained LLMs like Qwens. This approach could unify molecular generation with broader language-based interfaces.
>
> We see significant promise in these directions and plan to explore them further in future work.
>
> [1] Deng, Chaorui, et al. "Emerging properties in unified multimodal pretraining." arXiv:2505.14683 (2025).
>
> [2] Chen, Jiuhai, et al. "Blip3-o: A family of fully open unified multimodal models-architecture, training and dataset." arXiv:2505.09568 (2025).
>
>
>
> ---
> >Q3: Are the interpolation results robust across different dimensions of variation (molecular types, etc)?
>
> Thank you for the insightful question regarding interpolation robustness. We would appreciate further clarification on what is meant by "molecular types".  Does it refer to categories like peptides, antibodies, or other biomolecules? They are not in our current training set.
>
> In our study, we perform interpolation experiments across diverse molecules with different sizes and structures (see Figure 4, Table 4, and Table 5). These experiments demonstrate that our fixed-length latent space enables relatively smooth transitions in both structure and properties, indicating the robustness and regularity of the latent representation.
>
> If you could provide further clarification on "molecular types," we would be happy to elaborate on our results in more detail.
>
> ---

---

### Author Response · Authors · 2025-08-05

We sincerely thank all reviewers for your thoughtful engagement with our work and your constructive feedback. Your insights have helped us identify key areas for improvement and future exploration. Below we detail the concrete improvements we will make to the camera-ready version:
1. Additional Experiments
  - We will add comprehensive drug-likeness metrics (QED, SA, Lipinski) on GEOM-Drugs alongside validity/stability results, demonstrating MolFLAE’s superior performance in generating synthesizable, drug-like molecules.
  - We will report validity and stability metrics for all downstream tasks, confirming high validity and atom stability across manipulations.
2. Supplemental Revisions
  - Appendix D will be expanded to include concise, self-contained definitions of validity, atom stability, molecular stability, novelty, and all other evaluation criteria.
  - We will expand the related-work section to contrast MolFLAE’s fixed-length, permutation-invariant latent codes with the variable-length or residue-level latents representations used by prior works like UniMoMo and PepGLAD.
  - Finally, we will release a cleaned, fully reproducible codebase alongside camera-ready paper.

For reviewer-specific points not covered above, please see our individual rebuttals where we provide detailed responses.

These revisions will strengthen the paper’s rigor while preserving its core contribution: a unified, fixed-dimensional latent space enabling flexible 3D molecule manipulation beyond task-specific models. We are truly grateful for your expertise in shaping this work and are eager to continue discussing with you to further refine it.

---

### Note · Authors · 2025-08-12

We thank the AC and all reviewers for their time, thoughtful feedback, and constructive discussions throughout the process. The reviewing process has helped us both clarify and strengthen our work.

Following Reviewer EbXx’s suggestion on drug-likeness evaluation, we conducted additional experiments on QED, synthetic accessibility (SA), Lipinski's rule of five, and strain energy. These results, presented in our rebuttal, show that **MolFLAE largely outperforms baseline methods on these drug-likeness metrics**, reinforcing its practical relevance for drug design. As for other questions raised by reviewers, we appreciate that Reviewers EbXx and fHNV are satisfied with our rebuttals, and we would be happy to explore more on the remaining questions in the future.

We believe MolFLAE makes a distinct contribution by introducing **the first fixed-dimensional, E(3)-equivariant latent space for 3D molecules** independent of atom count, enabling high fidelity unconditional generation and **zero-shot, structure- and property-coordinated manipulations** that go beyond task-specific training. In addition to its strong performance on benchmarks, the case study on human glucocorticoid receptor optimization also demonstrates the **real-world potential** of our model.

We thank the AC and reviewers again for their insightful suggestions, which have strengthened our work and guided future research. We believe that MolFLAE represents a meaningful step forward in molecular generative modeling and look forward to building on this work to further advance flexible and interpretable 3D molecule design

---

### Decision · Program_Chairs · 2025-09-17

**Decision:**

Accept (poster)

**Comment:**

All reviewers assigned positive scores for this paper, although two reviews are still in the "borderline" regime.
In all reviews, it was mentioned that this work is of significant novelty, that the experimental setup is convincing, and that this work is relevant for  designing equivariant molecular representations. In particular, one reviewer mentioned that  the experiment al result nicely demonstrate the effectiveness of the proposed model in drug design applications.

There were also some concerns, mainly about the potential of latent disentanglement,  the model's ability to capture chirality, and some similarity to prior work. In the rebuttal, however, the authors could address most of these concerns in a way that seemed to be plausible for the reviewers. I also think that most of the critical remarks have been addressed in a convincing way by the authors, and therefore I recommend acceptance of this paper.